# Transformers Provably Learn Algorithmic Solutions for Graph Connectivity, But Only with the Right Data

Qilin Ye [* 1 2]   Deqing Fu [* 1]   Robin Jia [1]   Vatsal Sharan [1]

## Abstract

Transformers often fail to learn generalizable algorithms, instead relying on brittle heuristics. Using graph connectivity as a testbed, we explain this phenomenon both theoretically and empirically. We consider a simplified Transformer architecture, the Disentangled Transformer, and prove that an $L$-layer model can compute connectivity in graphs with diameters up to $3^L$, implementing an algorithm equivalent to computing powers of the adjacency matrix. By analyzing training dynamics, we prove that whether the model learns this strategy hinges on whether most training instances are within this model capacity. Within-capacity graphs (diameter $\leq 3^L$) drive the learning of the algorithmic solution while beyond-capacity graphs drive the learning of a simple heuristic based on node degrees. Finally, we empirically show that our insights transfer to standard Transformers: restricting training data to stay within a model's capacity makes both standard and Disentangled Transformers learn the exact algorithm.

## 1 Introduction

Large language models (LLMs) based on the Transformer architecture have demonstrated remarkable capabilities, yet their success is often shadowed by failures on tasks that demand robust, algorithmic reasoning. A growing body of evidence shows that, instead of learning generalizable algorithms, these models frequently rely on brittle shortcuts and spurious correlations that exploit statistical cues in the training data (Niven & Kao, 2019; Geirhos et al., 2020; Tang et al., 2023; Yuan et al., 2024; Zhou et al., 2024d; Ye et al., 2025). This shortcut reliance contributes to poor out-of-distribution (OOD) generalization, brittleness under superficial input changes, and unreliability on multi-step reasoning tasks (Zou et al., 2023; Deng et al., 2024; Li et al., 2024; Mirzadeh et al., 2025). On deterministic tasks, including shift ciphers, LLMs have been observed to favor high-probability outputs over correct solutions (McCoy et al., 2023), and in mathematical problem solving, strong in-distribution performance can fail to transfer when the problem structure or size changes (Saxton et al., 2019; Kao et al., 2024; Zhou et al., 2024b). Together, these findings motivate a basic question:

### When and why do Transformers learn heuristics over verifiably correct algorithms, even when the task admits an algorithmic solution?

We study this question using graph connectivity as a testbed. Connectivity is a fundamental problem in computational complexity (Wigderson, 1992), and it has three features that make it suitable for our analysis. First, it has a clear algorithmic solution. Reachability is the transitive closure of the graph, and it can be computed by matrix powering (Warshall, 1962; Floyd, 1962). Second, recent theory shows that Transformers with depth $L = \Theta(\log n)$ can express this matrix powering algorithm (Merrill & Sabharwal, 2025). Third, connectivity also admits simple heuristics, one example being degree counting. Informally speaking, for a randomly generated graph, two vertices with high degrees are more likely to be connected, but clearly degrees alone do not fully determine connectivity.

This setup leads to two distinct questions.

(Q1) Whether a Transformer is expressive enough to represent the algorithmic solution on a given graph instance.

(Q2) Whether gradient descent leads the model to this intended solution, or instead to a heuristic supported by the training distribution.

As a preliminary experiment, we answer (Q2) in the negative under standard training. We observe that Transformers can achieve perfect in-distribution accuracy on randomly generated Erdős-Rényi graphs, while failing catastrophically on simple OOD instances such as graphs consisting of two disjoint chains (see §3.3 and Fig. 1).

To systematically address both questions, we analyze *Disen-*

---

*Equal contribution  [1]Thomas Lord Department of Computer Science, University of Southern California [2]Department of Computer Science, Duke University. Correspondence to: Qilin Ye <qilin.ye@duke.edu>, Deqing Fu <deqingfu@usc.edu>.

*Proceedings of the 43rd International Conference on Machine Learning*, Seoul, South Korea. PMLR 306, 2026. Copyright 2026 by the author(s).

*tangled Transformers* (Friedman et al., 2023; Nichani et al., 2024), a simplified architecture that is more amenable to theoretical analysis while preserving the essential computations. We summarize our contributions below.

**An $L$-layer Disentangled Transformer solves connectivity up to diameter $3^L$, but no further.** We prove tight bounds that characterize the capacity of an $L$-layer Disentangled Transformer in terms of graph diameter. Let the diameter of a graph denote the maximum shortest-path distance between any two connected nodes (Def. 4.2). We show that an $L$-layer model is expressive enough to solve connectivity on every graph with diameter $\leq 3^L$ by implementing a matrix powering algorithm (Thm. 4.3). We complement this with a matching upper bound: Under mild assumptions, any $L$-layer model fails on some graph diameter $3^L + 1$ (Thm. 4.5). Together, these results establish that $3^L$ is the maximum diameter an $L$-layer model can handle perfectly. We call this quantity the model's *capacity*, and we empirically validate this diameter-depth scaling on both standard (Fig. 9) and Disentangled Transformers (Fig. 2).

**The computation separates into algorithmic and heuristic components.** We prove that, under the permutation symmetry induced by relabeling graph vertices, the weights of a trained Disentangled Transformer decompose into two distinct components, which we call *channels* (Thm. 4.7). The *algorithmic channel* simulates the matrix-powering algorithm using its hidden states across layers. The *heuristic channel*, on the other hand, ignores graph structure and instead computes global statistics based on node degrees, predicting connectivity from whether two nodes both have high degree. We also verify empirically that trained models approximately satisfy the required symmetry, making this decomposition applicable to practical settings (§4.3).

**Training dynamics select between these components based on the data distribution.** For Disentangled Transformers, our analysis of the training dynamics reveals a dichotomy driven by the data distribution. When the training distribution is dominated by within-capacity graphs, i.e., graphs with diameter $\leq 3^L$, the algorithmic channel is essentially able to handle the connectivity task by itself, rendering the heuristic channel redundant. Consequently, gradients suppress the heuristic channel and encourage the model to learn the matrix-powering algorithm (Thm. C.5). In contrast, when the distribution contains "many" beyond-capacity graphs, i.e., diameter $> 3^L$, the algorithmic channel cannot detect all connected pairs by itself. Consequently, the model attempts to make up for this by promoting the degree-based heuristic channel (Thm. C.9). In doing so, the model eventually learns a mixture of algorithmic computation and heuristics. Note that this characterization hinges on our exact $3^L$ capacity bound; an asymptotic one, such as the $\mathcal{O}(\exp(L))$ result from Merrill & Sabharwal (2025),

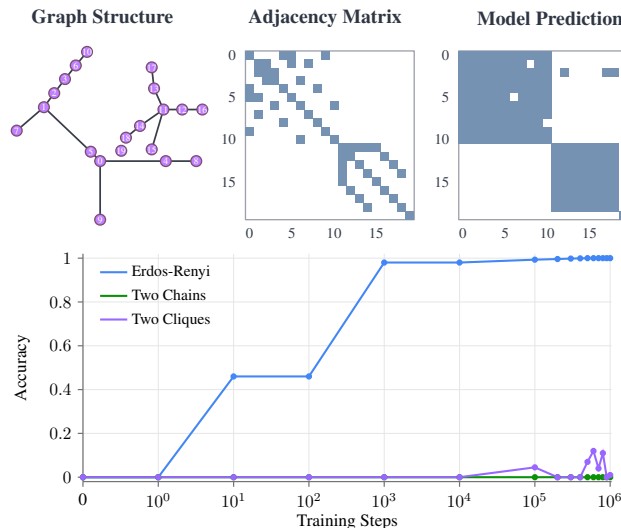

**Graph Structure**     **Adjacency Matrix**     **Model Prediction**

*Figure 1.* We train 2-layer Transformer models on Erdős-Rényi graphs. (**Top**) Visualization of a graph, its input adjacency matrix, and the model's (mis-)prediction of its connectivity. (**Bottom**) Although trained models predict connectivity nearly perfectly in-distribution, they fail to generalize to out-of-distribution graphs such as graphs with two isolated chains or cliques.

would not give the same criterion.

**Transformers (Disentangled *or not*) learn algorithmic solution *with the right data*.** Our theoretical insights suggest that restricting training to graphs within the model's capacity can make the models generalize better. In §5, we first verify this on Disentangled Transformers, by showing that the restriction promotes the algorithmic channel and suppresses the heuristic channel (Fig. 4). In §5.3, we further show that **our findings transfer empirically to standard Transformers.** We show that standard $L$-layer Transformers also have capacity $3^L$ and that they generalize *substantially* better to OOD graph families (Fig. 6) when trained on random graphs restricted to this diameter threshold.

Taken together, our results separate what the architecture can represent from what training selects. We show that Transformers can represent the matrix-powering solution for graph connectivity up to an exact diameter threshold, but training does not necessarily find this solution from arbitrary random graph data. Graphs beyond the model's capacity encourage the model to resort to heuristics, while graphs within capacity facilitate true algorithmic computation. Thus, for graph connectivity, the training distribution matters not only through its size or diversity, but also through how it aligns with the model's capacity.

## 2  Related Work

Theoretical work has studied both the expressive power and the limitations of Transformers. While Transformers are universal approximators for continuous sequence-to-sequence

functions (Yun et al., 2020), they also face sharper limits under more structured computational models. Fixed-depth attention has difficulty with periodic and hierarchical patterns (Hahn, 2020), standard Transformers are restricted to the complexity class $\mathsf{TC}^0$ under log-precision assumptions (Merrill & Sabharwal, 2023), and hard-attention variants are also confined to low-level circuit classes (Hao et al., 2022; Barcelo et al., 2024). Allowing depth to scale logarithmically with input length changes this picture, as log-depth Transformers can solve graph connectivity (Sanford et al., 2024; Merrill & Sabharwal, 2025). Chain-of-thought can also increase expressivity (Merrill & Sabharwal, 2024; Feng et al., 2023) but we exclude it here to focus only on what the base architecture learns through gradient descent.

Our work is also related to studies of algorithm learning in Transformers. Programmatic abstractions like RASP characterize algorithms that Transformers can implement and help study length generalization (Weiss et al., 2021; Zhou et al., 2024a). Empirically, for graph connectivity, Fu et al. (2024b) show that frontier LLMs can perform well on small graphs, while Saparov et al. (2025) show that Transformers struggle as graph size increases. Our analysis identifies graph diameter as the relevant capacity measure and studies when training selects the matrix-powering solution. This view is analogous to message passing in graph neural networks, where $L$ layers aggregate information from $L$-hop neighborhoods (Hamilton, 2020); in our setting, attention expands the reachable range to $3^L$. We include additional related work in the appendix §E.

## 3 Problem Setup and Preliminary Study

### 3.1 Graph Connectivity Task

We consider finite, undirected graphs on a fixed set of vertices $V = \{v_1, \ldots, v_n\}$. The model receives an adjacency matrix as input and must predict which pairs of vertices lie in the same connected component.

**Definition 3.1** (Self-loop Augmented **A**djacency Matrix). Let $G = (V, E)$ be a graph with $n$ vertices. We define the **self-loop augmented adjacency matrix** $A_G \in \{0, 1\}^{n \times n}$ as $A_{i,j} = 1$ if $\{v_i, v_j\} \in E$ or $i = j$, and 0 otherwise.

In other words, $A_G$ is the standard adjacency matrix with the identity matrix added. A key consequence is that the $(i, j)$-th entry of the matrix power $(A_G)^k$ counts the number of walks of length $k$ from $v_i$ to $v_j$. Since self-loops are included, a walk may stay at the same vertex for one or more steps. Throughout the paper, "adjacency matrix" refers to this self-loop augmented version unless stated otherwise.

**Definition 3.2** (Reachability Matrix). For a graph $G = (V, E)$, define its **reachability matrix** $R_G \in \{0, 1\}^{n \times n}$ as follows: $(R_G)_{i,j} = 1$ if there is a path between $v_i$ and $v_j$, and $(R_G)_{i,j} = 0$ otherwise. In particular, observe

$(R_G)_{i,j} = 1$ if and only if $(A_G)^n_{i,j} > 0$, as any two connected vertices are connected by a path of length at most $n - 1$.

When the context is clear, we drop the dependency of the underlying graph $G$ and simply write $A$ for adjacency and $R$ for connectivity matrix.

**The Objective.** Our objective is to learn a model $\mathcal{M} : \{0, 1\}^{n \times n} \to \mathbb{R}^{n \times n}$. Given adjacency matrix $A_G$, the model outputs a score matrix $\mathcal{M}(A_G)$, where positive scores are interpreted as positive connectivity predictions. We say that $\mathcal{M}$ is **perfect** on $G$ if, for every pair $(i, j)$ of vertices, $[\mathcal{M}(A_G)]_{i,j} > 0$ if and only if $(R_G)_{i,j} = 1$. We train on Erdős-Rényi graphs $\mathrm{ER}(n, p)$, where each possible edge on $n$ vertices is included independently with probability $p$.

### 3.2 Transformer Architectures

We first define the standard Transformer model used in the preliminary study. Let $A = A_G$ denote the self-loop augmented adjacency matrix of $G$. Each row of $A_G$ is treated as a token corresponding to one vertex, and each column is a feature tied to a vertex.

**Definition 3.3** (Transformers for Graph Connectivity). Fix depth $L$ and hidden width $d > n$. Let the linear read-in and read-out maps be

$$\mathsf{ReadIn}(X) := XW_{\mathrm{in}}, \quad W_{\mathrm{in}} \in \mathbb{R}^{n \times d},$$
$$\mathsf{ReadOut}(H) := HW_{\mathrm{out}}^\top, \quad W_{\mathrm{out}} \in \mathbb{R}^{n \times d}.$$

An $L$-layer single-head transformer for graph connectivity is defined as

$$\mathsf{TF}_\Theta^L(A) := \mathsf{ReadOut}\Big(\mathsf{Transformer}^L\big(\mathsf{ReadIn}(A)\big)\Big)$$

where $\mathsf{Transformer}^L$ is a standard pre-norm Transformer with self-attention and with *no* causal attention masks. We do not add separate positional encoding, since the diagonal entries in $A_G$ already provide node-identity information. A full specification is given in Definition A.1.

### 3.3 Preliminary Study

We train 2-layer Transformer models on $\mathsf{ER}(n = 20, p = 0.08)$ graphs and evaluate them on two OOD graph families: (1) 2Chain($n = 20, k = 10$) graphs with $n$ nodes consisting of two isolated chains each with $k$ nodes, and (2) 2Clique($n = 20, k = 10$) graphs with $n$ nodes consisting of two isolated $k$-Cliques. We measure the performance of model $\mathcal{M}$ via an exact match accuracy on our graph distribution $\mathcal{G}$, i.e., the probability that the model is perfect on the entire graph, mathematically defined as

$$\mathsf{ExactMatchAcc}(\mathcal{M}, \mathcal{G}) = \mathbb{E}_{G \in \mathcal{G}}\big[\mathcal{M} \text{ is perfect on } G\big].$$

**Result: Transformers fail to generalize.** As shown in Figure 1, the 2-layer Transformer model reaches almost perfect

exact-match accuracy on held-out graphs from the training distribution. However, its exact match accuracy is nearly zero on the 2Chain and 2Clique distributions. Thus, high in-distribution accuracy does not imply that the model has learned a connectivity algorithm that transfers to these OOD graph families. We observe the same qualitative behavior after extensive hyperparameter search and when increasing the number of layers. This motivates our theoretical analysis in §4 to investigate why transformers prefer to learn brittle heuristics and how we can encourage them to learn algorithmic solutions instead, and to separate what the architecture can represent from what gradient-based training learns.

# 4 Theory

We now present the main theoretical analysis. In §4.1, we define the Disentangled Transformer, the architectural proxy we analyze. Then, in §4.2, we prove Disentangled Transformer's exact capacity for graph connectivity in terms of graph diameters. Finally, in §4.3, we study the training dynamics and reveal how the model chooses between learning the intended algorithmic computation versus resorting to degree-based heuristics.

## 4.1 Disentangled Transformer

Standard Transformers are difficult to analyze directly because residual connections mix the outputs of different attention and MLP blocks. We instead analyze the Disentangled Transformer (Friedman et al., 2023; Nichani et al., 2024). In this architecture, MLPs are removed and each attention block appends its output as a new coordinate slice of the residual stream rather than adding it to the existing representation. The hidden dimension grows with depth, but the computation due to each layer remains easily tractable.

Previous work has shown that the Disentangled Transformer can serve as a reasonable proxy for standard Transformers. Nichani et al. (2024) show that any standard attention-only Transformer can be re-expressed as a disentangled model by specializing attention to implement feature concatenation. Chen et al. (2024) also use this architecture because it preserves the computations of interest while making the residual stream easier to analyze. We now formalize it.

**Definition 4.1** (Disentangled Transformer). Let $G$ be a graph on $n$ nodes with self-loop augmented adjacency matrix $A$. Fix a depth $L$. For each layer $\ell = 1, \cdots, L$, let the hidden dimension be $d_\ell = 2^{\ell+1} \cdot n$, and let $W_\ell \in \mathbb{R}^{d_{\ell-1} \times d_{\ell-1}}$ be the attention parameter matrix. Finally, let $W_O = [I_n, \cdots, I_n] \in \mathbb{R}^{n \times d_L}$ be the fixed read-out matrix, and let $\Theta = \{W_\ell\}_{\ell=1}^L$ denote the trainable weights. The $L$-layer Disentangled Transformer $\mathsf{TF}_\Theta^L$ maps $A$ to an $n \times n$ score matrix as follows,

**Input hidden state** $h_0 = [I_n, A] \in \mathbb{R}^{n \times d_0}$

**$\ell$-th hidden state** $h_\ell = [h_{\ell-1}, \mathrm{Attn}(h_{\ell-1}; W_\ell)] \in \mathbb{R}^{n \times d_\ell}$

**Model output** $\mathsf{TF}_\Theta^L(A) = h_L W_O^\top$

where $\mathrm{Attn}(h_{\ell-1}; W_\ell) = \frac{1}{n} \cdot \mathsf{ReLU}(h_{\ell-1} W_\ell h_{\ell-1}^\top) h_{\ell-1}$. Thus, each layer doubles the number of $n$-dimensional feature blocks. In particular, $h_\ell$ is a "flat" matrix with dimension $n \times d_\ell$ (and recall $d_\ell = 2^{\ell+1} \cdot n$).

## 4.2 Expressivity and Capacity

We now study what the Disentangled Transformer can represent on the graph connectivity task. Since the input adjacency matrix includes self-loops, powers of $A$ encode reachability at increasing path lengths, as $[A^k]_{i,j} > 0$ if and only if there is a walk from $v_i$ to $v_j$ of length at most $k$. Thus, a natural way to solve connectivity is to compute sufficiently high powers of $A$. The question is how far this computation can propagate in $L$ layers.

We show that the answer is exact: An $L$-layer Disentangled Transformer can solve connectivity on all graphs whose connected components have diameter at most $3^L$. Under mild assumptions, this is also the largest diameter for which perfect prediction can be guaranteed.

**Definition 4.2** (Graph Diameter). Let $G = (V, E)$ be a finite, simple, undirected graph. For $u, v \in V$, we let $d_G(u, v)$ be the *shortest-path distance* between $u, v$, which is finite if they are connected and infinite otherwise.

For a connected component $C$ of $G$, we define its *diameter* to be the maximum shortest-path distance between any two vertices in $C$. We define the diameter of $G$, denoted $\mathrm{diam}(G)$, as the maximum diameter over all connected components of $G$. Under this convention, $\mathrm{diam}(G)$ is finite even when $G$ is disconnected.

We first give the expressivity result, in which we prove that the model can implement the matrix-powering needed for connectivity, up to a distance determined by its depth.

**Theorem 4.3** (Lower Bound: Expressivity). *There exists an $L$-layer Disentangled Transformer $\mathsf{TF}_\Theta^L$ that makes perfect predictions for every graph $G$ satisfying $\mathrm{diam}(G) \leq 3^L$.*

*Sketch of proof.* For all $\ell$, setting $W_\ell = I_{d_{\ell-1}}$ suffices. These choices of weights implement the matrix powering algorithm $\sum_{j=0}^{3^L} \alpha_j A^j$ with positive coefficients $\alpha_j$. □

The previous theorem gives a constructive lower bound on what depth L can achieve. We next show that, for nonnegative weights, this bound is tight.

**Definition 4.4** (Model Capacity). The **capacity** of an $L$-layer Disentangled Transformer $\mathsf{TF}_\Theta^L$ is the largest integer $d$ such that there exist weights achieving perfect predictions on every graph $G$ with $\mathrm{diam}(G) \leq d$.

**Theorem 4.5** (Upper Bound: Capacity). *Let $\mathsf{TF}_\Theta^L$ be an $L$-layer Disentangled Transformer on $n = \Omega(3^L)$ nodes.*

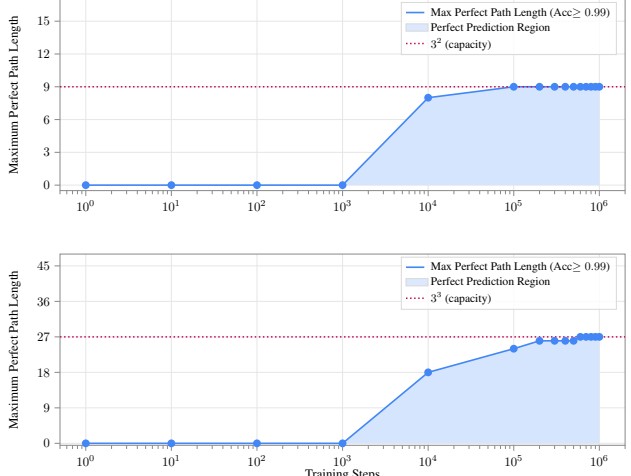

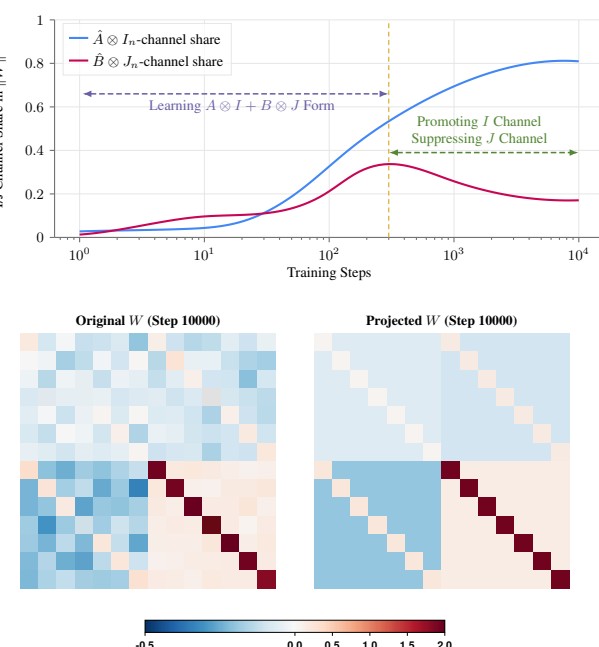

*Figure 2*. **Capacity of Disentangled Transformers.** We train 2-layer (**top**) and 3-layer (**bottom**) Disentangled Transformers on $\mathrm{ER}(n = 24)$ and $\mathrm{ER}(n = 64)$ graphs respectively. When evaluated on hold-out sets, both models can only make reliable predictions ($\geq 99\%$ accuracy) on node pairs $u, v$ if and only if $d_G(u, v) \leq 3^L$. These findings resonate with our theoretical observations in Theorem 4.5.

*Assume that the weights $W_\ell \geq 0$ entrywise for each $\ell$. Then there exists a graph $G$ with $\mathrm{diam}(G) = 3^L + 1$ on which $\mathrm{TF}_\Theta^L$ is not perfect. Therefore, under nonnegative weights, no $L$-layer Disentangled Transformer can guarantee perfect connectivity prediction beyond diameter $3^L$.*

We remark that the nonnegative assumption is used in the upper bound proof. However, in §5.1, we show that the same $3^L$ diameter threshold appears empirically, even when nonnegativity is not enforced during training.

*Sketch of proof.* We prove the claim using a dichotomy. If some intermediate attention score creates a false positive across disconnected components (Lemma B.1), we use a backtracking argument to preserve that false positive while adding a path of length $3^L + 1$. Otherwise, no false positives ever occur (Lemma B.2). In this case, we use an induction over layers to show that "information" regarding connectivity can spread only to distance $3^\ell$ by layer $\ell$. Thus, a depth-$L$ model cannot recognize a connected pair at distance $3^L + 1$. For the full proof, see Section B.2. □

This $3^L$ bound will be used throughout the training-dynamics analysis. We introduce the following terminology.

**Definition 4.6** (Within-capacity and Beyond-capacity Pairs at Depth $L$). Fix a graph $G$ with self-loop augmented adjacency matrix $A$, and fix a depth $L$. A pair of vertices $(i, j)$ is **within capacity** if $[A^{3^L}]_{i,j} > 0$ and **beyond capacity** otherwise. In other words, a pair $(i, j)$ is within capacity if and only if they are connected and have a shortest distance $\leq 3^L$.

*Figure 3*. **Training Dynamics of Disentangled Transformers.** We train a 1-layer Disentangled Transformer on graphs from $\mathrm{ER}(n = 8, p = 0.2)$ distribution. Weight $W$ approximately approaches the $A \otimes I_n + B \otimes J_n$ form. (**Top**) There are two major phases during training, where during Phase 1, model focuses on learning the equivariant parameterizations so both $I$ and $J$ channel's share of energy (see §5.1) in $W$ increases, and during Phase 2, the algorithmic $I$-channel is promoted and the heuristic $J$-channel is suppressed. (**Bottom**) Visualization of the learned weights and their projection to the closest $\hat{W} = \hat{A} \otimes I_n + \hat{B} \otimes J_n$ form.

Overloading the notation, we also say $G$ is a within-capacity graph if $\mathrm{diam}(G) \leq 3^L$ and beyond-capacity otherwise.

### 4.3  Training Dynamics

The capacity threshold from the previous section gives a characterization on the Disentangled Transformer's theoretical limits. We now study how this property affects training. The analysis proceeds in two steps. First, we identify a parameterization that decomposes the model weights into an algorithmic channel and a heuristic one. Then, we analyze how these two components interact with each other under gradient descent induced by different training samples.

**Parameterizing Model Weights.** The graph connectivity problem is invariant under relabeling of vertices. If a graph with adjacency matrix $A$ has connectivity matrix $R$, then the relabeled graph $PAP^\top$ has connectivity matrix $PRP^\top$ for any permutation matrix $P$. We also observe empirically that Disentangled Transformers trained from random initialization quickly become approximately layer-wise equivariant under these relabelings (Figure 7). This motivates analyzing training inside the subspace of layer-wise permutation-equivariant weights. The next theorem

characterizes this subspace via equation 1.

**Theorem 4.7** (Layerwise Permutation-Equivariant Parameterization). *Suppose an $L$-layer Disentangled Transformer $\mathsf{TF}_\Theta^L$ has non-negative weights. Let $K_{\ell-1} = 2^\ell$. Then $\mathsf{TF}_\Theta^L$ is layer-wise permutation equivariant, i.e., for each $\ell$, any hidden states $h \in \mathbb{R}^{n \times d_{\ell-1}}$, and any permutation $P \in S_n$,*

$$\mathrm{Attn}(Ph(I_{K_{\ell-1}} \otimes P^\top); W_\ell) = P \, \mathrm{Attn}(h; W_\ell) \, (I_{K_{\ell-1}} \otimes P^\top),$$

*if and only if each layer weight $W_\ell$ admits a decomposition*

$$W_\ell = A_\ell \otimes I_n + B_\ell \otimes J_n \qquad (1)$$

*for some $A_\ell, B_\ell \in \mathbb{R}^{2^\ell \times 2^\ell}$ for all $\ell$, where $\otimes$ denotes the Kronecker product and $J_n = \mathbf{1}\mathbf{1}^T$ the all-ones $(n \times n)$ matrix.*

*Sketch of proof.* Sufficiency is immediate, as both $I_n$ and $J_n$ are invariant under conjugation by permutation matrices.

For the converse, layerwise equivariance of the bilinear attention map forces $(I_K \otimes P)W_\ell(I_K \otimes P^\top) = W_\ell$ for every permutation $P$, and it can be shown that this implies every $n \times n$ subblock of $W_\ell$ commutes with all permutations. This implies each subblock lies in $\mathrm{span}\{I_n, J_n\}$. See the end of Appendix C.1 for the full proof. $\square$

Observe that this parameterization contains the constructive lower bound used by the matrix-powering algorithm in Theorem 4.3. In fact, it is also compatible with the capacity upper bound, as Theorem C.2 shows that for any model that "reaches" the capacity, the symmetric part of the weights that drive attention must lie in the $I_n$-channel. Finally, the population gradient stays inside this algebra (Theorem C.4), so that we can analyze training as an interaction between just the two channels. We now describe their functions.

**The $I_n$-channel ($A_\ell \otimes I_n$) implements the matrix-powering algorithm.** The term $A_\ell \otimes I_n$ preserves locality inside each block. When applied within the attention mechanism, it only combines features between nodes that share graph neighbors. Across layers, this channel composes multi-hop information by effectively computing powers of the adjacency matrix $A$. After $L$ layers, the readout aggregates a weighted sum $\sum_{j=0}^{3^L} \alpha_j A^j$ with nonnegative coefficients, which is the matrix-powering computation used in Thm. 4.3.

**The $J_n$-channel ($B_\ell \otimes J_n$) collects heuristics.** The term $B_\ell \otimes J_n$ broadcasts information globally, ignoring graph structure. Indeed, since $J_n x = (\mathbf{1}^\top x)\mathbf{1}$, it sums a feature over all nodes and broadcasts the result. In particular, $A J_n = \mathbf{d}\mathbf{1}^\top$ where $\mathbf{d} = A\mathbf{1}$ is the degree vector. Thus, this channel can compute statistics based on node degrees and related global quantities. These statistics can be predictive on random graphs, but they do not determine reachability and can fail on adversarial graph families.

Appendices C.3 and C.4 make these interpretations formal by analyzing the channel derivatives and sample gradients.

**Training Dynamics.** We now analyze gradient descent in this two-channel parameterization. The analysis is population-level and uses a nonnegative parameterization so that the model remains monotone, and the ReLU does not introduce sign changes.

**Assumption 4.8.** We use the following assumptions for the training-dynamics analysis.

(i) **Data Distribution.** Graphs are sampled from $\mathsf{ER}(n, p)$, i.e., Erdős-Rényi graphs with $n$ vertices and each edge independently included with probability $p \in (0, 1)$. We assume disconnected graphs occur with positive probability.

(ii) **Nonnegative Equivariant Parameterization.** For each layer $\ell$, we assume the weight $W_\ell \geq 0$ is parameterized as $W_\ell = A_\ell \otimes I_n + B_\ell \otimes J_n$ with entrywise nonnegative $A_\ell, B_\ell$.

(iii) **Surrogate Loss for the Analysis.** Given model scores $Z = \mathsf{TF}_\Theta^L(\cdot) \in \mathbb{R}_{\geq 0}^{n \times n}$, we convert them into probability-like values using the monotone map[1] $\phi(z) = 1 - e^{-\alpha z}$ with $\alpha > 0$, and apply entrywise cross-entropy with respect to the connectivity matrix $R$. We choose this surrogate because our correctness criterion is binary (depending only on whether $Z > 0$), while the training dynamics argument requires a differentiable objective. The map $\phi$ preserves this distinction and yields a simple closed form gradient. Formally, the loss is

$$\mathcal{L}(Z; R) = -\sum_{i,j} \big( R_{i,j} \log \phi(Z_{i,j}) + (1 - R_{i,j}) \log(1 - \phi(Z_{i,j})) \big),$$

and its gradient with respect to $Z$ is

$$\frac{\partial \mathcal{L}}{\partial Z} = \alpha \left(1 - R/\phi(Z)\right) \in \mathbb{R}^{n \times n},$$

where division is defined entrywise.

We remark that the parameterization in (ii) is used for the theoretical dynamics analysis. In §5.1, we train Disentangled Transformers without enforcing this form and observe that the learned weights nevertheless approach the $A \otimes I_n + B \otimes J_n$ structure.

Under these assumptions, we can characterize both convergence and the structure of limiting points of gradient descent. Recall $\Theta$ denotes the set of trainable weights (Definition 4.1). With (ii) in place, this reduces to just the channel parameters $(A_\ell, B_\ell)_{\ell=1}^L$.

---

[1]It is possible that $R_{i,j} = 1$ while $Z_{i,j} = 0$, in which case $\partial \mathcal{L}/\partial Z$ is undefined. To avoid this technical issue, the formal analysis uses $\phi_\epsilon = 1 - (1 - \epsilon)e^{-\alpha z}$ for some small $\epsilon > 0$. All subsequent analyses hold verbatim by replacing $\phi$ with $\phi_\epsilon$.

**Theorem 4.9** (Convergence to KKT Points). *Let* $\mathcal{R}(\Theta) = \mathbb{E}_{G \sim \mathsf{ER}(n,p)}[\mathcal{L}(\mathsf{TF}_\Theta^L(A_G); R_G)]$ *denote the population risk. For* $\lambda > 0$, *define the regularized objective* $\mathcal{R}_\lambda(\Theta) = \mathcal{R}(\Theta) + \frac{\lambda}{2}\|\Theta\|_F^2$. *We optimize over constraint set* $\mathcal{C} = \{(A_\ell, B_\ell)_\ell : A_\ell \geq 0, B_\ell \geq 0, \forall \ell\}$.

*Consider the sequence* $\{\Theta^{(k)}\}_{k \geq 0}$ *generated by projected gradient descent on* $\mathcal{R}_\lambda$:

$$\Theta^{(k+1)} = \Pi_\mathcal{C}\left(\Theta^{(k)} - \eta \nabla \mathcal{R}_\lambda(\Theta^{(k)})\right), \qquad (2)$$

*with sufficiently small step size* $\eta > 0$ *and initialization* $\Theta^{(0)} \in \mathcal{C}$ *of the form* $W_\ell = A_\ell \otimes I + B_\ell \otimes J$. *Then every limit point* $\Theta_\lambda^* \in \mathcal{C}$ *satisfies the KKT conditions:*

$$\nabla_{B_\ell}\mathcal{R}(\Theta_\lambda^*) + \lambda B_\ell^* \geq 0, \quad B_\ell^* \geq 0, \\ (\nabla_{B_\ell}\mathcal{R}(\Theta_\lambda^*) + \lambda B_\ell^*) \odot B_\ell^* = 0, \qquad (3)$$

*and analogously for* $A_\ell^*$. *Moreover, the iterates converge to a KKT point at the standard* $\mathcal{O}(1/\epsilon)$ *rate for projected gradient descent.*

The proof follows the standard convergence analysis for projected gradient descent on smooth, nonconvex functions over a closed, convex set (Bertsekas, 1997; Beck, 2017). We defer the details to Appendix C.5.

Theorem 4.9 reduces the training dynamics question to the structure of KKT points. We now ask when such a point can keep a nonzero heuristic channel. In Appendix C.3 we systematically study how the objective changes in the presence of the $J_n$-channel. Informally speaking, this directional derivative compares two effects. On one hand, for pairs of vertices in different connected components (i.e. the pairs that are disconnected), the $J_n$-channel can mistakenly raise scores for labels that should be zero, so the loss wants to penalize (downweight) the $J_n$-channel. On the other hand, for pairs within the same connected component, the $J_n$-channel can raise scores for labels that should be 1, and that aligns with the objective, so the loss wants to reward (upweight) the $J_n$-channel.[2] Therefore, the heuristic channel can remain only if the second effect compensates for the first.

**Theorem 4.10** (Learning the Algorithm, informal version of Theorem C.5 and Corollary C.20). *If the penalty from raising scores on disconnected pairs (thus creating false positives) outweighs the reward from raising scores on connected pairs (true positives), then a limiting KKT point cannot retain the global heuristic channel. Consequently, the model converges to learning the fully algorithmic matrix-powering algorithm.*

The formal statements are in Theorem C.5 for the unregularized population risk and Corollary C.20 for the regularized

---

[2]There are important qualifiers here (not all connected pairs reward the $J_n$-channel, and not all disconnected pairs penalize it), but we omit them to streamline the flow. The details can be found in Theorem C.9.

objective. With convergence guaranteed by Theorem 4.9, the training process can be understood through two phases.

**Phase 1: Both channels pick up "easy" examples**. Early in training, many connected pairs are still underpredicted (connectivity matrix has $R_{i,j} = 1$, yet the score $Z_{i,j}$ is far from it). The local $I_n$-channel can improve these pairs by composing neighborhood information, and the global $J_n$-channel can also raise their scores through graph-level heuristics. At this stage, the reward from connected pairs far outweigh the penalty from disconnected pairs, so both channels quickly ramp up. Phase 1 is transient and ends once those "easy" connected pairs are mostly saturated. In Figure 3 (top), it only occupies around $2 \cdot 10^2$ steps out of $10^4$ total.

**Phase 2: The data distribution determines which channel wins**. After the "easy" connected pairs are mostly saturated, the reward to the $J_n$-channel weakens. The remaining behavior is governed by the population-level balance, described in Theorem C.5. Disconnected pairs penalize the $J_n$-channel when it creates cross-component (false) positives, but connected pairs reward the same channel when it raises scores on true positives.

**Final Outcome.** There are two potential outcomes, and they depend on the within-capacity and beyond-capacity distinction from Definition 4.6. First, if the training distribution is dominated by within-capacity graphs ($\mathrm{diam}(G) \leq 3^L$), then the $I_n$-channel *alone* can account for connected pairs using the local matrix-powering computation. In some sense, the $J_n$-channel becomes redundant and essentially only receives penalty if the model predicts false positives, so gradient descent suppresses it, leaving the model to only learn the algorithmic $I_n$-channel in the long run.

In contrast, if the training distribution contains a significant share of beyond-capacity graphs ($\mathrm{diam}(G) > 3^L$), then the algorithmic $I_n$-channel cannot solve the task on its own. Graphs on which $I_n$ becomes insufficient therefore reward the $J_n$-channel instead. In the long run, both channels coexist, and the model learns a mixture of the matrix-powering algorithm and the degree-counting heuristic.

In more technical details, Theorem C.9 gives the sample-level dynamics behind both cases, and Remark C.13 gives the corresponding population-level mixture. To sum up, the final learned solution is determined by how well the training distribution aligns with the $3^L$ capacity threshold, namely whether the $I_n$-channel suffices for the task by itself. If yes, then the model learns a pure algorithmic solution; if no, then an additional heuristic is learned along the way. In Figure 5, we visualize this transition by varying the proportion of beyond-capacity graphs in the training distribution. This criterion motivates the data restriction in the next section.

# 5 Experiments

We now validate the theoretical picture empirically. The theory makes two structural assumptions that are useful for the analysis: The capacity upper bound (Theorem 4.5) assumes nonnegative weights, and the training dynamics analysis assumes the nonnegative equivariant parameterization (Assumption 4.8 (ii)). In the experiments, we do *not* impose these constraints unless stated otherwise.

First, in §5.1, we show that trained Disentangled Transformers follow the predicted $3^L$ diameter threshold (Theorems 4.3 and 4.5) and that they naturally approach the algorithmic-heuristic decomposition (Assumption 4.8 (ii)). Then, in §5.2, we show that restricting Disentangled Transformer training to within-capacity graphs promotes the matrix-powering solution, while preserving at-capacity examples ($\mathrm{diam}(G) = 3^L$) is also important for generalization. Finally, in §5.3, we return to standard Transformers and show that our key results on the $3^L$ capacity and on training data prescription transfer empirically.

## 5.1 Capacity and Training Dynamics

**$L$-layer Disentangled Transformers Hit Their Capacity at Exactly $3^L$.** We train 2-layer and 3-layer Disentangled Transformers on Erdős-Rényi graphs with 24 or 64 nodes respectively[3]. As shown in Figure 2, the trained models make reliable predictions, with at least $99\%$ accuracy, on node pairs whose shortest-path distance $\leq 3^L$, while predictions beyond this threshold dip significantly. These results match the exact capacity bound of Disentangled Transformers in Theorem 4.5 and support the within-capacity versus beyond-capacity distinction in Definition 4.6. We also note that although the capacity theorem (Theorem 4.5) assumes nonnegative weights, we do not enforce it during training. Yet, the predicted diameter threshold remains accurate.

**Disentangled Transformers Learn an Algorithm-Heuristic Mixture.** We next examine the training dynamics of Disentangled Transformers. We train a 1-layer model on $\mathrm{ER}(n=8, p=0.2)$ graphs, randomly initialized and without enforcing any parameterization constraint. As shown in Figure 3, the learned weight matrix $W$ becomes close to the form $A \otimes I_n + B \otimes J_n$ for some matrices $A, B \in \mathbb{R}^{2 \times 2}$. Deeper models show the same behavior, as shown in Figure 8. These results show the applicability of the decomposition in Theorem 4.7.

Then, to quantify the two channels, we project the learned weight $W$ onto this algebra by finding the closest matrix $\hat{W} = \hat{A} \otimes I_n + \hat{B} \otimes J_n$ in Frobenius norm. We then track the channel shares defined in §D. During training, the

---

[3]The number of nodes is chosen so that $n > 2 \cdot 3^L$. This enables us to easily test the model's performance on the two-chains distribution, where each chain has length $> 3^L$.

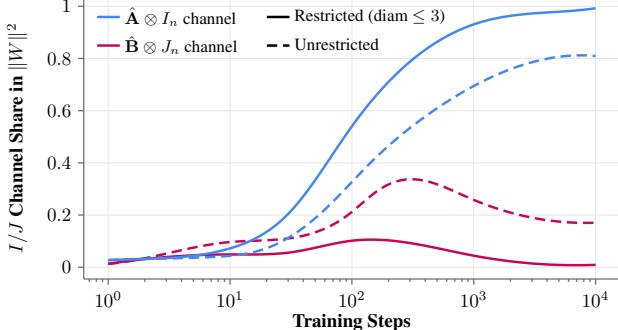

*Figure 4.* Following insights from Theorems C.5 and C.9, we repeat the same experiment setup as in Figure 3 but only training on within-capacity graphs (see Definition 4.6). As shown in the **solid** lines, restricting training samples by capacity pushes the energy share of the **algorithmic** mechanism (the $A \otimes I_n$ channel) further to nearly 100% in the weight $W$. It simultaneously prevents the growth of the **heuristic** portion (the $B \otimes J_n$ channel).

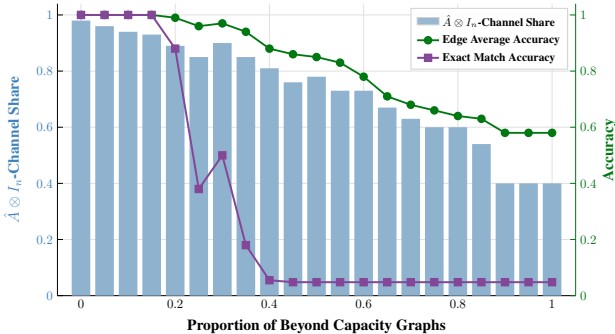

*Figure 5.* We vary the proportion of beyond-capacity graphs, and train the same Disentangled Transformer on stratified ER distribution and test on the same OOD 2Chain distribution. We find that Transformers are robust towards a small amount of noise (beyond-capacity graphs). Although the $W$ is not exactly in the $A \otimes I_n$ form, the model still performs perfectly when the energy share of $I$-channel dominates (beyond roughly 90%).

share of the algorithmic channel $\hat{A} \otimes I_n$ increases, while the share of the heuristic channel $\hat{B} \otimes J_n$ first increases and then decreases. This provides empirical evidence for the two-phase story in §4.3: Both channels can grow early in training, while the heuristic channel is later suppressed when the data favors the matrix-powering computation.

## 5.2 Encouraging Transformers to Learn Algorithms

**Restricting to within-capacity graphs promotes the algorithmic channel.** Our theory suggests a direct change to the training distribution: remove graphs that lie beyond the model's capacity. For a graph distribution $\mathcal{G}$, we define its within-capacity part as $\mathcal{G}_{\leq} = \{G \in \mathcal{G} : \mathrm{diam}(G) \leq 3^L\}$, and let $\mathcal{G}_{>}$ denote the remaining graphs. In Figure 4, we repeat the 1-layer Disentangled Transformer from Figure 3, but train only within-capacity graphs from $\mathrm{ER}_{\leq}$. This restriction promotes the algorithmic $\hat{A} \otimes I_n$ channel and

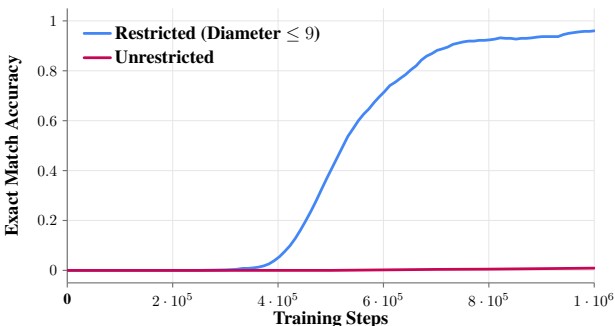

*Figure 6.* **Standard Transformer models learn generalizable solutions from within capacity data.** We train a 2-layer standard Transformer model on $\mathsf{ER}(n = 20)$ graphs, with and without restricting graph diameters to $3^2 = 9$. When tested on OOD 2Chain graphs, the one trained with *the right data* is able to generalize.

suppresses the heuristic $\hat{B} \otimes J_n$ one. Eventually, the learned weight is almost entirely in the algorithmic channel.

**At-capacity graphs are important.** The restriction removes beyond-capacity graphs, but it should not remove the hardest graphs within this capacity. For a 1-layer Disentangled Transformer, the capacity is 3. In Figures 11 and 12, we vary a cutoff $d$ by restricting the training graphs to satisfy $\mathrm{diam}(G) \le d$, and evaluate on 2Chain graphs. When $d < 3$, the model learns to solve shorter paths but fails to generalize to longer within-capacity paths. When $d > 3$, the training distribution includes beyond-capacity graphs, and the model again struggles to learn the algorithmic solution. The best behavior occurs at $d = 3$, which excludes beyond-capacity graphs but preserves at-capacity examples.

**A small amount of beyond-capacity data can be tolerated.** We next vary how much beyond-capacity structure appears in the training distribution. Define

$$\rho(\mathcal{G}) = \mathbb{E}_{G \in \mathcal{G}} \frac{|\{(u, v) \in V, d_G(u, v) > 3^L\}|}{n^2}$$

to be the expected fraction of beyond-capacity node pairs in a graph from $\mathcal{G}$. In practice, we control this quantity by stratified sampling from the mixture $\mathcal{G}_q = q\mathcal{G}_\le + (1-q)\mathcal{G}_>$.

As shown in Figure 5, the model remains robust in this experiment when $\rho$ is small. In this regime, the model maintains high energy share in the algorithmic channel and makes perfect predictions on OOD 2Chain graphs. As $\rho$ increases, the algorithmic-channel share decreases, and OOD performance undergoes a phase change where it quickly drops to near zero. Our results suggest an empirical tolerance level $\rho^* > 0$, below which the model can still rely on the algorithmic channel for prediction.

### 5.3 Transfer to Standard Transformers

**Standard Transformers admit the same empirical capacity.** Our theory is mainly developed for Disentangled

Transformers, which we use as an architectural proxy for standard Transformers due to their simplicity. Nevertheless, we now close the loop by showing that the key results transfer apply to standard Transformers as well. As shown in Figure 9, a 2-layer standard Transformer exhibits the same capacity behavior, where predictions are reliable within the $3^2$ threshold and unreliable beyond it. This does not directly extend Theorem 4.5 to standard Transformers, but it supports using the same capacity as an empirical guide.

**The same restriction improves OOD generalization on standard Transformers.** We then apply the within-capacity restriction to the standard Transformer model from §3.3. We train the same 2-layer standard Transformer on $\mathsf{ER}(n = 20, p = 0.08)$ graphs, but restrict the training distribution to graphs with diameter at most 9. As shown in Figure 6, the model trained on the restricted distribution generalizes to OOD 2Chain graphs, while the model trained on the unrestricted Erdős-Rényi distribution does not. The same restriction also improves generalization to OOD 2Clique graphs, as shown in Figure 10.

These results show that the prescription from the Disentangled Transformer analysis transfers empirically to standard Transformers. The standard Transformer exhibits the same $3^L$ capacity bound, and restricting training to within-capacity graphs improves OOD generalization on both 2Chain and 2Clique graphs. Although the theoretical mechanism is established for Disentangled Transformers, the same capacity-based training set restriction is effective for the standard architecture used in the preliminary study.

## 6 Conclusion

In this paper, we studied how Transformers learn graph connectivity by separating what the architecture can represent from what training selects. For Disentangled Transformers, we proved that an $L$-layer model can implement the matrix-powering solution on graphs with $\mathrm{diam}(G) \le 3^L$, and that this threshold is tight under the nonnegative-weight assumption. We then used this capacity threshold to explain training behavior: Within-capacity data favors the algorithmic channel, while beyond-capacity structure may reward the degree-based heuristic channel. Our experiments show that strategically changing the training distribution can change the learned computation. For Disentangled Transformers, restricting training to within-capacity graphs suppresses the heuristic channel, promotes the matrix-powering solution, and improves OOD generalization. For standard Transformers, we observe the same empirical capacity behavior and show that the same within-capacity restriction improves OOD generalization. Overall, our results show that for graph connectivity, learning the intended algorithm depends on how the training distribution aligns with the model's capacity.

## Impact Statement

This paper presents work whose goal is to advance the field of machine learning, specifically our understanding of when neural networks learn generalizable algorithms versus brittle heuristics. Our findings have potential implications for improving the reliability and interpretability of machine learning systems deployed in safety-critical applications, where algorithmic correctness is paramount. By identifying data distribution properties that encourage algorithmic learning, this work may inform better training practices. We do not foresee direct negative societal consequences from this foundational research.

## Acknowledgments

The authors acknowledge the Center for Advanced Research Computing (CARC) at the University of Southern California for providing computing resources that have contributed to the research results reported within this publication. We also acknowledge the use of the USC NLP cluster provided by USC NLP Group. This work used the Delta system at the National Center for Supercomputing Applications through allocation CIS250737 from the Advanced Cyberinfrastructure Coordination Ecosystem: Services & Support (ACCESS) program, which is supported by National Science Foundation grants #2138259, #2138286, #2138307, #2137603, and #2138296. DF and RJ were also supported by gifts from the USC-Capital One Center for Responsible AI and Decision Making in Finance (CREDIF) and the USC-Amazon Center on Secure and Trusted Machine Learning. RJ was also supported by the National Science Foundation under Grant No. IIS-2403436. VS was supported by an NSF CAREER Award CCF-2239265, an Amazon Research Award, a Google Research Scholar Award and an Okawa Foundation Research Grant. The work was done in part while DF and VS were visiting the Simons Institute for the Theory of Computing. Any opinions, findings, and conclusions or recommendations expressed in this material are those of the author(s) and do not reflect the views of the funding agencies.

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

# Appendix

## A   Additional Details on Problem Setups and Preliminary Studies

**Definition A.1** (Transformer for Graph Connectivity: full specification)**.**

**Input and output**. Given a simple graph on $n$ nodes with adjacency matrix $A \in \{0,1\}^{n \times n}$, let $\bar{A} = A + I_n$ be its self-loop augmented adjacency matrix. We treat $\bar{A}$ as input embedding: row $i$ is the token for node $i$; column $j$ indexes a feature tied to node $j$. To ease notation, we will simply write $A$ in place of $\bar{A}$ and always assume the adjacency matrix is self-loop augmented. The model outputs an $n \times n$ score matrix $\text{TF}_\Theta^L(A)$, the predicted connectivity matrix.

**Dimensions and parameters**. Fix depth $L$, hidden dimension $d > n$, number of heads $H$ with $d = H d_h$, and feed-forward width $d_{\text{ff}}$. The parameters we need include:

$$W_{\text{in}}, W_{\text{out}} \in \mathbb{R}^{n \times d}, \qquad W_{\ell,h}^Q, W_{\ell,h}^K, W_{\ell,h}^V \in \mathbb{R}^{d \times d_h}, \qquad W_\ell^O \in \mathbb{R}^{H d_h \times d}$$

$$W_\ell^{(1)} \in \mathbb{R}^{d \times d_{\text{ff}}},\, b_\ell^{(1)} \in \mathbb{R}^{d_{\text{ff}}}, \qquad W_\ell^{(2)} \in \mathbb{R}^{d_{\text{ff}} \times d},\, b_\ell^{(2)} \in \mathbb{R}^d$$

for $\ell = 1, \ldots, L$ and heads $h = 1, \ldots, H$. We use pre-norm residual blocks with LayerNorm (LN) and GeLU activations. We do not use attention masks or any extra positional encoding; the identity in $\overline{A}$ already pins each token to a node.

**The forward map**. The read-in is linear: $h^{(0)} = \overline{A} W_{\text{in}} \in \mathbb{R}^{n \times d}$. From there, for each $\ell = 1, \ldots, L$, let $\tilde{h} = \text{LN}(h^{(\ell-1)})$ as we use pre-norm. Within each block:

$$\textbf{Multi-head self-attention} \qquad Q_{\ell,h} = \tilde{h} W_{\ell,h}^Q, \quad K_{\ell,h} = \tilde{h} W_{\ell,h}^K, \quad V_{\ell,h} = \tilde{h} W_{\ell,h}^V$$

$$\textbf{Attention scores} \qquad \alpha_{\ell,h} = \frac{1}{n} \text{ReLU}(1/\sqrt{d_h} \cdot Q_{\ell,h} K_{\ell,h}^\top), \quad z_{\ell,h} = \alpha_{\ell,h} V_{\ell,h}$$

$$\textbf{Concatenation \& residual} \qquad z_\ell = [z_{\ell,1} \mid \ldots \mid z_{\ell,H}] W_\ell^O \in \mathbb{R}^{n \times d}, \qquad u_\ell = h^{(\ell-1)} + z_\ell$$

$$\textbf{Feed-forward} \qquad \hat{u}_\ell = \text{LN}(u_\ell), \qquad \text{FFN}_\ell(\hat{u}_\ell) = \text{GeLU}(\hat{u}_\ell W_\ell^{(1)} + b_\ell^{(1)}) W_\ell^{(2)} + b_\ell^{(2)}$$

and finally $h^{(\ell)} = u_\ell + \text{FFN}_\ell(\hat{u}_\ell) \in \mathbb{R}^{n \times d}$. The read-out is linear: $\text{TF}_\Theta^L(A) = h^{(L)} W_{\text{out}}^\top \in \mathbb{R}^{n \times n}$.

**Metrics for Permutation Equivariance.** Let $P \in S_n$ be the corresponding permutation matrix for any $\sigma \in \mathcal{S}_n$. For a given graph adjacency matrix $A$, we compute the model's prediction in respect to $P_\sigma$ as $\mathcal{M}(P_\sigma A P_\sigma^\top)$. Now we define a equivariance consistency metric, **Equivariance Consistency via Frobenius Cosine Similarity**:

$$\text{ConsFrob}(\mathcal{M}) = \mathbb{E}_{\sigma \in \mathcal{S}_n, A \in \mathcal{G}} \left[ \frac{\langle \mathcal{M}(P_\sigma A P_\sigma^\top), P_\sigma \mathcal{M}(A) P_\sigma^\top \rangle_F}{\|\mathcal{M}(P_\sigma A P_\sigma^\top)\|_F \|P_\sigma \mathcal{M}(A) P_\sigma^\top\|_F} \right] \tag{4}$$

When measuring intermediate model computations, this metric is modified depending on the model type. For standard Transformer models, $\mathcal{M}^\ell$ computes the final Readout to the hidden states at layer $\ell$. For Disentangled Transformers we are computing $P_\sigma \mathcal{M}^\ell(A)(P_\sigma \otimes I_n)^\top$.

# B Details for Capacity

## B.1 Expressivity

**Theorem 4.3.** *There exists an L-layer Disentangled Transformer that makes perfect predictions for every graph $G$ satisfying* $\mathrm{diam}(G) \leq 3^L$.

*Proof.* Set $W_\ell = I_{d_{\ell-1}}$ for all layers and note that all matrices are entrywise nonnegative, so ReLU and the factor $1/n$ never changes supports. With $h_0 = [I \mid A] = [A^0 \mid A^1]$ and update $h_\ell = [h_{\ell-1} \mid (h_{\ell-1}h_{\ell-1}^\top)h_{\ell-1}/n]$, we can show by induction that every $n \times n$ block of $h_\ell$ lies in $\mathrm{span}\{A^0, \ldots, A^{3^\ell}\}$, and that some block contains $A^{3^\ell}$ with a positive coefficient. Indeed, the base case holds trivially; for the inductive step, if a block within $h_{\ell-1}$ contains $A^m$, then $(h_{\ell-1}h_{\ell-1}^\top)h_{\ell-1}$ contains $A^{2m}A^m = A^{3m}$. Finally, the readout simply sums over all these blocks, so $\mathrm{supp}(\mathsf{TF}_\Theta^L(A)) = \mathrm{supp}(A^{3^L})$.

Finally, because $A$ has self-loops, supports are monotone in power and stabilizes at $t \geq \mathrm{diam}(G)$. Thus, if $\mathrm{diam}(G) \leq 3^L$ we get $\mathrm{supp}(\mathsf{TF}_\Theta^L(A)) = \mathrm{supp}(A^{\mathrm{diam}(G)})$. $\qquad\square$

## B.2 Capacity

**Theorem 4.5.** *Fix $L \geq 1$ and let $\mathsf{TF}_\Theta^L$ be an L-layer Disentangled Transformer on $n = \Omega(3^L)$ nodes. Further assume that the weights $W_\ell \geq 0$ for each $\ell$. Then there exists a graph $G$ with diameter of $3^L + 1$ on which $\mathsf{TF}_\Theta^L(A)$ is not perfect. In other words, diameter $3^L$ upper bounds the capacity of any L-layer Disentangled Transformer. In particular, taking $n \geq (7/3) \cdot 3^L + 2$ suffices.*

For each layer $\ell$ we define the post-ReLU score $R_\ell = \mathrm{ReLU}(h_{\ell-1}W_\ell h_{\ell-1}^\top)$. The proof of the theorem will be partitioned into two branches: whether some intermediate $R_\ell$ gives a false positive on some graph, or all $R_\ell$'s are free of false positives on all graphs. We say a pair of nodes $(u, v)$ from $G$ is a *witness* to false positives if they belong to different connected components while $R_\ell(G)_{u,v} > 0$. Throughout this section, we set $n \geq (7/3) \cdot 3^L + 2$.

**Lemma B.1.** *Assume the setup in Theorem 4.5. Suppose there exist some $n$-node graph, a layer index $\ell^* \in \{1, \ldots, L\}$, and vertices $u, v$ belonging to different connected components of the graph such that $(R_{\ell^*})_{u,v} > 0$. Further assume that $\ell^*$ is globally minimal, in the sense that for all $n$-node graphs and all $\ell < \ell^*$, the corresponding $R_\ell$ has no false positive entries across components. Then there exists a graph $G$ such that $\mathrm{diam}(G) = 3^L + 1$, $\mathsf{TF}_\Theta^L(A(G))_{u,v} > 0$, where $u, v$ lie in different connected components of $G$.*

*Proof.* The proof roughly partitions into two parts. In the first half, we backtrack the computation DAG, tracing the "sources" that contribute to the false positiveness of $(u, v)$. This gives us subgraphs which we call *certificates* that, if kept untouched, suffice to guarantee a false positiveness of $(u, v)$. In the second half, we construct a graph $G$ that preserves these certificates while also containing a path of length $> 3^L$ disjoint from both certificates, and we show that $\mathsf{TF}_\Theta^L$ preserves false positiveness of $(u, v)$ on $G$, thereby proving the claim.

STEP 1. CONSTRUCTING THE CERTIFICATES. Intuitively, since $R_{\ell^*}(H)_{u,v} > 0$, there exist column indices $p, q$ with $(W_{\ell^*})_{p,q} > 0$, $h_{\ell^*-1}(H)_{u,p} > 0$, and $h_{\ell^*-1}(H)_{v,q} > 0$. We will backtrack the entries that contribute to the positiveness of the hidden states entries $h_{\ell^*-1}(H)_{u,p}$ and $h_{\ell^*-1}(H)_{v,q} > 0$ in the computation DAG, iteratively visiting previous layers. Formally, we define a *certificate* for an entry $h_t(H)_{i,c} > 0$ to be a small tree whose nodes are triples [of form (layer, row, column)] recording earlier entries that must be positive to guarantee that the current one is positive. The root is $(t, i, c)$ and we build it top-down by repeating one of the two rules until we hit the first layer, which we know looks like $[I_n \mid A]$. We now describe how to backtrack. Since $h_t = [h_{t-1} \mid \mathrm{Attn}(h_{t-1}; W_t)]$, we split the recursion on layer $t$ into two cases: whether the entry lies in the first half ($h_{t-1}$) or the second half ($\mathrm{Attn}(h_{t-1}; W_t)$).

- (First half) If column $c$ is in the inherited block of $h_t$, add a single child $(t-1, i, c)$ to $(t, i, c)$, as the value is simply copied from the previous layer $h_{t-1}$.

- (Second half) If column $c$ is in the newly appended block, then by definition

$$h_t(H)_{i,c} = \frac{1}{n}\sum_k R_t(H)_{i,k}h_{t-1}(H)_{k,c'} \qquad \text{for some } c',$$

and since this is a sum of nonnegative terms, there exists at least one $k$ with $R_t(H)_{i,k} > 0$ and $h_{t-1}(H)_{k,c'} > 0$. In turn,

$$R_t(H)_{i,k} = \sum_{r,s} h_{t-1}(H)_{i,r} (W_t)_{r,s} h_{t-1}(H)_{k,s} > 0,$$

which implies that there exist indices $r, s$ with $(W_t)_{r,s} > 0$, $h_{t-1}(H)_{i,r} > 0$, and $h_{t-1}(H)_{k,s} > 0$. Thus, for such $(t, i, c)$, we create three children:

$$(t - 1, k, c'), \qquad (t - 1, i, r), \qquad (t - 1, k, s).$$

Now let $s(t)$ denote the maximal number of vertices needed to realize a single certificate for some entry $h_t(\cdot) > 0$ by the recursive procedure above. At $t = 0$ we may assume $s(0) \le 2$. The recursion gives $s(t) \le 3s(t - 1)$, so $s(t) \le 2 \cdot 3^t$. Since $u, v$ lie in different connected components of $H$, and $\ell^*$ is minimal, every index $k$ selected by a certificate at any layer $t \le \ell^* - 1$ stays within the same component as its $i$, so the two certificates induce trees $T_u, T_v$ that occupy disjoint vertex sets $S_u, S_v$, with $|S_u \cup S_v| \le 4 \cdot 3^{\ell^* - 1} \le 4 \cdot 3^{L-1}$ vertices.

STEP 2. BUILDING A NEW GRAPH. Initialize $G$ to the edgeless graph, keeping node isolated. We then embed $T_u, T_v$ onto $G$ by adding edges according to the trees. Finally, we connect $3^L + 2$ vertices outside $S_u \cup S_v$ arbitrarily into a long chain of path length $3^L + 1$. This is always possible because there are $n - |S_u \cup S_v|$ vertices outside the union, which is $\ge \big((7/3) \cdot 3^L + 2\big) - 4 \cdot 3^{L-1} = 3^L + 2$, and this is why require $n \ge (7/3) \cdot 3^L + 2$ in this Lemma.

We claim $G$ is the graph we seek. On one hand, every sum used by the certificates is a sum of nonnegative terms, and we have preserved a strictly positive summand at each step that appears in the tree. Hence $R_{\ell^*}(G)_{u,v} > 0$ with $u, v$ also disconnected in $G$. On the other hand, under the choice of $n$ specified by Theorem 4.5, there exist at least $3^L + 2$ vertices outside $S_u \cup S_v$, so connecting them into a long path guarantees $\mathrm{diam}(G) = 3^L + 1$. The claim then follows. $\qquad\square$

**Lemma B.2.** *Assume the setup in Theorem 4.5. Further assume that for every $n$-node graph $G$ and every layer $\ell \in \{1, \dots, L\}$, the post-ReLU scores $R_\ell(G)$ has no positive entry between distinct connected components of $G$. Then, for every graph $G$ and every $u, v \in V(G)$, if $\mathsf{TF}_\Theta^L(A(G))_{u,v} > 0$, we must have $\mathrm{dist}_G(u, v) \le 3^L$. Consequently, if $G$ contains a connected component of diameter $3^L + 1$ then $\mathsf{TF}_\Theta^L$ is not perfect on $G$.*

*Proof.* Under the no-false-positives assumption, the idea is to show that "information" spreads no faster than power base 3 so $\mathsf{TF}_\Theta^L$ never predicts "Yes" on node pairs with distance beyond $3^L$. Concretely, columns exchange information as attention scores are calculated. We first define the "distances" between columns by giving each column a label $\in \{1, \dots, n\}$, and then show that by layer $\ell$, two columns can "share" information if and only if their labels, *interpreted as graph nodes,* are within distance $3^\ell$.

STEP 1. GIVING EACH COLUMN A LABEL. We first consider trivial graph $G_0$ with $n$ isolated nodes: immediately $h_0(G_0) = [I_n \mid I_n]$ and, by hypothesis, every $R_\ell(G_0)$ have no off-diagonal positives. Inductively this shows that every column of $h_\ell(G_0)$ has support in exactly one row. We define the label of this column to be the row index $\in \{1, \dots, n\}$ where the unique support is. With labels defined, the remaining proof is based on establishing the following locality claim.

CLAIM. Fix graph $G$, layer $\ell$, and $i, j \in \{1, \dots, n\}$. If column $c$ of $h_\ell(G)$ has label $j$ and if $h_\ell(G)_{i,c} > 0$, then $\mathrm{dist}_G(i, j) \le 3^\ell$. In other words, *every column spreads at most $3^\ell$ hops away from its label by depth $\ell$.*

STEP 2. ESTABLISHING THE CLAIM. We prove this claim via induction. The base case $\ell = 0$ directly follows from the fact that $h_0(G) = [I_n \mid A(G)]$. For the inductive step, we assume that the claim holds at depth $\ell - 1$ with radius $3^{\ell-1}$. As in Lemma B.1, there are two column types in $h_\ell$: inherited or newly appended columns. The former case is easy; if $c$ is inherited from $h_{\ell-1}$, then $h_\ell(G)_{i,c} = h_{\ell-1}(G)_{i,c}$, so the bound follows from the inductive hypothesis. We now assume $c$ is newly appended.

Suppose $(R_\ell(G) h_{\ell-1}(G))_{i,c} > 0$ for a column $c$ with label $j$. Then there exists a row $k$ with $R_\ell(G)_{i,k} > 0$ and $h_{\ell-1}(G)_{k,c} > 0$. By the IH, $\mathrm{dist}_G(k, j) \le 3^{\ell-1}$. Then we expand $R_\ell(G)_{i,k} > 0$ to obtain column witnesses $p, q$, with $h_{\ell-1}(G)_{i,p} > 0$, $h_{\ell-1}(G)_{k,q} > 0$, and $(W_\ell)_{p,q} > 0$, as in Lemma B.1. Let $a, b$ be the labels of $p, q$, respectively. By IH again, $\mathrm{dist}_G(i, a) \le 3^{\ell-1}$ and $\mathrm{dist}_G(k, b) \le 3^{\ell-1}$. We now split the analysis into two cases.

- If $a \ne b$, we derive a contradiction to the no-false-positives assumption by reusing the certificate procedure from Lemma B.1. Because $W_\ell \ge 0$ entrywise, every positive entry in $h_t(\cdot)$ admits a certificate supported on at most $s(t) \le 2 \cdot 3^t$ vertices. In particular, there exist certificates witnessing $h_{\ell-1}(G)_{i,p} > 0$ (labeled $a$) and $h_{\ell-1}(G)_{k,q} > 0$

(labeled $b$). Let $S_a, S_b$ be the corresponding certificate vertex sets. Form a new graph $G'$ on the same $n$ vertices whose connected components are two disjoint induced copies $S'_a, S'_b$ of the subgraphs on $S_a, S_b$ (leaving all other vertices outside $S_a \cup S_b$ isolated). Note this is feasible because $|S_a| + |S_b| \leq 4 \cdot 3^{L-1} \leq n$ assumed by Theorem 4.5. By construction, there exist $i' \in S'_a$ and $k' \in S'_b$ with $h_{\ell-1}(G')_{i',p} > 0$ and $h_{\ell-1}(G')_{k',q} > 0$. Thus,

$$(h_{\ell-1}(G')W_\ell h_{\ell-1}(G')^\top)_{i',k'} \geq h_{\ell-1}(G')_{i',p}(W_\ell)_{p,q}h_{\ell-1}(G')_{k',q} > 0,$$

meaning $R_\ell(G')_{i',k'} > 0$. But $i', k'$ belong to different connected components in $G'$, contradiction! Therefore,

- $a = b$. Triangle inequality gives $\text{dist}_G(i,j) \leq \text{dist}_G(i,a) + \text{dist}_G(a,k) + \text{dist}_G(k,j) \leq 3 \cdot 3^{\ell-1} = 3^\ell$, completing the induction.  END PROOF OF CLAIM / STEP 2.

The model's output $\mathsf{TF}_\Theta^L(A(G)) = h_L(G)W_O^\top$ is an entrywise nonnegative sum over the $n \times n$ blocks of $h_L(G)$. Since each block respects the $3^L$ locality bound, we have $\mathsf{TF}_\Theta^L(A(G))_{u,v} = 0$ whenever $\text{dist}_G(u,v) \geq 3^L + 1$. Hence, on any graph whose largest component has diameter $3^L + 1$, the model will inevitably miss a pair $(u,v)$ of nodes realizing this diameter. $\square$

*Proof of Theorem 4.5.* Combine Lemmas B.1 and B.2. $\square$

# C  Details for Training Dynamics

## C.1  Characterizing Block Weights $W_\ell$

As discussed in §4.3, due to the symmetric nature of the graph connectivity problem, it is natural to demand that a "good" model should map not only adjacency matrices $A$ to connectivity matrices $R$, but also $PAP^\top$ to $PRP^\top$ for any permutation $P$. We further generalize equivariance. Observe that given a permutation matrix $P$ and any hidden states $h \in \mathbb{R}^{n \times (kn)}$ consisting of $k$ consecutive $n \times n$, the mapping $h \mapsto Ph(I_K \otimes P^\top)$ relabels both rows and columns within each $n \times n$ block in a way that is consistent with the effects of $P$. Hence, the notion of equivariance can be generalized to any (nonnegative) hidden states, beyond just the ones induced by adjacency matrices.

Similarly, we are now also able to define an $L$-layer Disentangled Transformer on arbitrary inputs of appropriate dimensions. For any nonnegative initial state $h_0 \in \mathbb{R}^{n \times 2n}$, recursively define $h_\ell = [h_{\ell-1} \mid \text{Attn}(h_{\ell-1}; W_\ell)]$ for $\ell = 1, \ldots, L$. Let $\text{Sum}(h)$ denote the sum of the consecutive left-aligned $n \times n$ blocks of $h$. Then the generalized output is $\mathsf{TF}_\Theta^L(h_0) = \text{Sum}(h_L)$. We define two equivariance-related conditions. The first one is a direct generalization of $P\mathsf{TF}_\Theta^L(A)P^\top = \mathsf{TF}_\Theta^L(PAP^\top)$; the second one, as discussed in §4.3, makes theoretical analysis significantly more tractable while also being supported by empirical evidence.

**Definition C.1** (Output Equivariance and Layerwise Attention Equivariance)**.** Let $\mathsf{TF}_\Theta^L$ be an $L$-layer Disentangled Transformer with nonnegative weights. Let $K_\ell = 2^{\ell+1}$.

(i) For $h_0 \in \mathbb{R}_{\geq 0}^{n \times 2n}$ and for any $P$, define $h_0^P = Ph_0(I_{K_0} \otimes P^\top)$. We say $\mathsf{TF}_\Theta^L$ is **output-level value equivariant** iff $P\mathsf{TF}_\Theta^L(h_0)P^\top = \mathsf{TF}_\Theta^L(h_0^P)$ holds for all $P$ and all $h_0 \in \mathbb{R}_{\geq 0}^{n \times 2n}$.

(ii) We say $\mathsf{TF}_\Theta^L$ is **layer-wise attention equivariant** iff for each $\ell$ and any hidden states $h \in \mathbb{R}^{n \times d_{\ell-1}}$ (i.e., any hidden states of dimension feasible for layer $\ell$),

$$\text{Attn}(Ph(I_{K_{\ell-1}} \otimes P^\top); W_\ell) = P\,\text{Attn}(h; W_\ell)\,(I_{K_{\ell-1}} \otimes P^\top),$$

**Theorem C.2** (Parameterization of "Good" Models)**.** *Let $n = \Omega(3^L)$ as in Theorem 4.5. Fix an $L$-layer Disentangled Transformer $\mathsf{TF}_\Theta^L$ with nonnegative weights. Suppose that*

*(i) $\mathsf{TF}_\Theta^L$ is output-level value-equivariant, and*

*(ii) $\mathsf{TF}_\Theta^L$ reaches its capacity bound of $3^L$, i.e., for every graph, we have $\text{supp}(\mathsf{TF}_\Theta^L(A)) = \text{supp}(A^{3^L})$.*

*Then, either $\mathsf{TF}_\Theta^L$ or a functionally equivalent version of it satisfies the following: for each layer $\ell$, there exists a nonnegative matrix $\Lambda_\ell \in \mathbb{R}^{K_{\ell-1} \times K_{\ell-1}}$ such that $W_\ell + W_\ell^\top = \Lambda_\ell \otimes I_n$. In other words, $W_\ell$ can be decomposed into this form up to an antisymmetric part.*

*(Note that the theorem is a direct generalization of equivariance under all graph permutations; replacing $h_0$ by $[I_n \mid A]$ gives the desired result for a fixed graph with adjacency matrix $A$.)*

*Proof.* To prove the claim, it suffices to show that if we partition $W_\ell$ into $K_{\ell-1} \times K_{\ell-1}$ contiguous sub-blocks of size $n \times n$, then each block must be diagonal, with symmetry conditions meeting $W_\ell + W_\ell^\top = \Lambda_\ell \otimes I_n$.

To do so, the proof is split into two parts: we prove that each $n \times n$ block must be diagonal using (ii) and Lemma B.2, and that the diagonal entries must realize the said forms by examining the forward maps under a curated, parameterized class of initial hidden states.

STEP 1: EACH BLOCK MUST BE DIAGONAL. In this step, we argue that if a block admits a positive off-diagonal entry, then the certificate trick from Lemma B.1 will create a false positive entry on some output, contradicting (ii).

Formally, let $R_\ell = \mathsf{ReLU}(h_{\ell-1}W_\ell h_{\ell-1}^\top)$. If for some graph and some $\ell$, there exists a false positive entry $(R_\ell)_{i,k} > 0$ for some $i, k$ across different connected components, then the false positiveness would persist to the output, contradicting (ii). Hence $\mathsf{TF}_\Theta^L$ must have no false positives.

Consider feeding the graph $G_0$ of $n$ isolated vertices into $\mathsf{TF}_\Theta^L$, so that $h_0(G_0) = [I_n \mid I_n]$. The premises of Lemma B.2 hold, so every column of every $h_\ell(G_0)$ is supported in exactly one row, which we called its label in $\{1, \ldots, n\}$. Hence, if we write $h_\ell(G_0) = [X_1^{(\ell)} \mid \ldots \mid X_{K_\ell}^{(\ell)}]$ of contiguous $n \times n$ blocks, then each such block $X_r^{(\ell)}$ must be nonnegative and diagonal. Now expand

$$R_\ell = \mathsf{ReLU}(h_{\ell-1}W_\ell h_{\ell-1}^\top) = h_{\ell-1}W_\ell h_{\ell-1}^\top = \sum_{r,s} X_r^{(\ell-1)} W_\ell[r,s](X_s^{(\ell-1)})^\top.$$

We first claim that every $n \times n$ sub-block $W_\ell[r,s]$ is diagonal. Suppose not, that there exist indices $r, s$ and distinct nodes $i \neq k$ such that $(W_\ell[r,s])_{i,k} > 0$. For a node $i$ and a block $r$, we say $(i,r)$ is *activatable* at depth $\ell - 1$ if there exists *some* graph $G$ such that $X_r^{(\ell-1)}(G)[i,i] = h_{\ell-1}[i,(r-1)n+i] > 0$. Two cases:

- If at least one of $(i,r)$ or $(k,s)$ is not activatable, then for every graph $G$, at least one factor $X_r^{(\ell-1)}(G)[i,i]$ or $X_s^{(\ell-1)}(G)[k,k]$ is zero, and thus $(W_\ell[r,s])_{i,k}$ is functionally inert and never contributes to any $R_\ell$ entry. Hence we may simply set it to 0 without altering the model's output on any graph.

- If both $(i,r)$ and $(k,s)$ are activatable, take graphs $G_i, G_k$ that make $X_r^{(\ell-1)}(G_i)[i,i] > 0$ and $X_x^{(\ell-1)}(G_k)[k,k] > 0$. Using the certificate mechanism in Lemma B.1, each positiveness admits a finite certificate subgraph with at most $2 \cdot 3^{\ell-1}$ vertices. We then create a new graph $G'$ and disjointly embed both certificates into it, leaving all other vertex isolated. The two labels $i, k$, viewed as nodes, now lie in different components. But then the product

$$X_r(G')[i,i] \cdot (W_\ell[r,s])_{i,k} \cdot X_k(G')[k,k] > 0,$$

making $(R_\ell(G'))_{i,k} > 0$, contradiction.

Therefore $W_\ell[r,s]$ is diagonal for all block indices $(r,s)$. This concludes STEP 1.

STEP 2. $W_\ell$ IS NODE-SYMMETRIC. Given a triplet $(\ell, r, s)$, we can now write $W_\ell[r,s]$ as $\mathrm{diag}(w_{\ell,r,s}(1), \ldots, w_{\ell,r,s}(n))$. Our goal is to show that for each $(\ell, r, s)$, $w_{\ell,r,s}(j) + w_{\ell,s,r}(j) = w_{\ell,r,s}(k) + w_{\ell,s,r}(k)$ for all $j, k \in [n]$. We formalize this in matrix form: For each node $i \in [n]$ and each layer $\ell$, let $\Lambda_\ell^{(i)} = [w_{\ell,r,s}(i)]_{r,s} \in \mathbb{R}^{K_{\ell-1} \times K_{\ell-1}}$ and define the symmetric part $\mathrm{Sym}(\Lambda_\ell^{(i)}) = (\Lambda_\ell^{(i)} + \Lambda_\ell^{(i)T})/2$; the goal is to show that given $\ell$, all $\Lambda_\ell^{(i)}$ are the same, so that $\mathrm{Sym}(W_\ell) = \Lambda_\ell \otimes I_n$ or equivalently, $W_\ell + W_\ell^\top = \Lambda_\ell \otimes I_n$, as claimed.

Throughout out this step, we will use a family of special hidden states parameterized by a scalar $\lambda > 0$ and a vector $u = (u_1, u_2) \in \mathbb{R}_{\geq 0}^2$. Fix distinct nodes $j \neq k$. For $\lambda, u$, define the initial state $h_0(\lambda, u) \in \mathbb{R}^{n \times 2n}$ by setting exactly four entries nonzero:

$$\begin{cases} h_0(\lambda,u)[j,j] = \lambda u_1 & h_0(\lambda,u)[j,n+j] = \lambda u_2 \\ h_0(\lambda,u)[k,k] = \lambda u_1 & h_0(\lambda,u)[k,n+k] = \lambda u_2. \end{cases}$$

Note that $h_0(\lambda, u)$ is invariant under the transposition $P = (j,k)$, i.e., $Ph_0(\lambda,u)(I_{K_0} \otimes P^\top) = h_0(\lambda, u)$. Therefore, by assumption (i), we must have $\mathsf{TF}_\Theta^L(h_0)_{j,j} = \mathsf{TF}_\Theta^L(h_0)_{k,k}$. Let $h_\ell(\lambda, u)$ be the network state at depth $\ell$. Because of STEP 1, there is no cross-row interaction for this input at any depth. Writing the row-$i$ vector as $v_\ell^{(i)}(\lambda, u) \in \mathbb{R}^{K_\ell}$, recursion gives, for $i \in \{j, k\}$,

$$v_0^{(i)}(\lambda, u) = \lambda u, \qquad v_\ell^{(i)}(\lambda, u) = [v_{\ell-1}^{(i)}(\lambda, u) \mid q_\ell^{(i)}(\lambda, u)v_{\ell-1}^{(i)}(\lambda, u)]$$

where

$$q_\ell^{(i)}(\lambda, u) = \frac{1}{n} \cdot v_{\ell-1}^{(i)}(\lambda, u)^\top \operatorname{Sym}(\Lambda_\ell^{(i)}) \, v_{\ell-1}^{(i)}(\lambda, u).$$

Taking $\ell_1$-norms gives

$$\|v_\ell^{(i)}(\lambda, u)\| = (1 + q_\ell^{(i)}(\lambda, u))\|v_{\ell-1}^{(i)}(\lambda, u)\| \quad \text{and} \quad \|v_L^{(i)}(\lambda, u)\| = \|v_0^{(i)}(\lambda, u)\| \prod_{\ell=1}^{L}(1 + q_\ell^{(i)}(\lambda, u)). \tag{5}$$

Because the readout weight $W_O$ is a concatenation of $I_n$'s, and under our specific input $h_0(\lambda, u)$, every nonzero row $i$ lies in columns with indices $i$ modulo $n$, the $(i, i)$ output numerically equals $\|v_L^{(i)}(\lambda, u)\|$. Hence, assumption (i) requires $\|v_L^{(j)}(\lambda, u)\| = \|v_L^{(k)}(\lambda, u)\|$.

Let $\ell^*$ be the minimal layer such that $\operatorname{Sym}(\Lambda_{\ell^*}^{(j)}) \neq \operatorname{Sym}(\Lambda_{\ell^*}^{(k)})$. If no such $\ell^*$ exists for all $j \neq k$, then all $\Lambda_\ell^{(i)}$'s are the same given any fixed $\ell$, and STEP 2 holds. Otherwise, for every $\ell < \ell^*$, the symmetric parts coincide, and $v_{\ell-1}^{(j)}(\lambda, u) = v_{\ell-1}^{(k)}(\lambda, u)$ and $q_\ell^{(j)}(\lambda, u) = q_\ell^{(k)}(\lambda, u)$ for all $\lambda, u$. We may use $v_{\ell^*-1}(\lambda, u)$ to denote both $v_{\ell^*-1}^{(j)}(\lambda, u)$ and $v_{\ell^*-1}^{(k)}(\lambda, u)$ for they are now equal.

Because of the structure of $h_0(\lambda, u)$, by induction, the row vectors of each hidden state admits an odd power expansion

$$v_{\ell-1}^{(i)}(\lambda, u) = \lambda u + \lambda^3 \xi_{1,\ell-1}(u) + \lambda^5 \xi_{2,\ell-1}(u) + \dots$$

from which we conclude $q_\ell^{(i)}(\lambda, u) = O(\lambda^2)$ for every $\ell$. In particular, at $\ell = \ell^*$,

$$q_{\ell^*}^{(j)}(\lambda, u) - q_{\ell^*}^{(k)}(\lambda, u) = \frac{1}{n} \cdot v_{\ell^*-1}(\lambda, u)^\top (\operatorname{Sym}(\Lambda_{\ell^*}^{(j)}) - \operatorname{Sym}(\Lambda_{\ell^*}^{(k)}))v_{\ell^*-1}(\lambda, u) = \lambda^{2m} c(u) + o(\lambda^{2m})$$

for some $m \geq 1$ and some nondegenerate polynomial $c(u)$ as $\lambda \searrow 0$. In particular,

$$q_{\ell^*}^{(j)}(\lambda, u) - q_{\ell^*}^{(k)}(\lambda, u) = \Theta(\lambda^{2m}).$$

We now put this back into the comparison between the output's $(j, j)$ and $(k, k)$ entry. Recall that $v_0^{(j)}(\lambda, u) = v_0^{(k)}(\lambda, u) = \lambda u$. Further, since $v_{\ell-1} = \lambda u + O(\lambda^3)$, we know $q_\ell(\lambda, u) = O(\lambda^2)$ for every $\ell$ and every $i$. We drop $\lambda, u$ for notational simplicity. It follows from equation 5 that

$$\|v_L^{(j)}\| - \|v_L^{(k)}\| = \lambda\|u\| \cdot \left[\prod_{l<\ell^*}(1 + q_\ell)\right] \cdot \left[\left[1 + q_{\ell^*}^{(j)}\right]\prod_{\ell>\ell^*}\left[1 + q_\ell^{(j)}\right] - \left[1 + q_{\ell^*}^{(k)}\right]\prod_{\ell>\ell^*}\left[1 + q_\ell^{(k)}\right]\right]$$

$$= \lambda\|u\| \cdot \left[\prod_{\ell<\ell^*}(1 + O(\lambda^2))\right] \cdot \left[q_{\ell^*}^{(j)} - q_{\ell^*}^{(k)}\right] \cdot \left[\prod_{\ell>\ell^*}(1 + O(\lambda^2))\right]$$

$$= \lambda\|u\|\Theta(\lambda^{2m})(1 + o(1)) = \Theta(\lambda^{2m+1})$$

which is nonzero for small $\lambda$. Hence the $(j, j)$ and $(k, k)$ entries can be made different, contradicting assumption (i), and the proof is complete! $\qquad\square$

**Theorem 4.7.** *Suppose an L-layer Disentangled Transformer $\mathsf{TF}_\Theta^L$ has nonnegative parameters. Suppose $\mathsf{TF}_\Theta^L$ is layerwise permutation equivariant, i.e., for each $\ell$, any hidden states $h \in \mathbb{R}^{n \times d_{\ell-1}}$, and any permutation $P \in S_n$,*

$$\operatorname{Attn}(Ph(I_{K_{\ell-1}} \otimes P^\top); W_\ell) = P\operatorname{Attn}(h; W_\ell)\,(I_{K_{\ell-1}} \otimes P^\top),$$

*then each block $W_\ell = A_\ell \otimes I_n + B_\ell \otimes J_n$ for some $A_\ell, B_\ell \in \mathbb{R}^{K_{\ell-1}, K_{\ell-1}}$. In other words, each block-aligned $n \times n$ submatrix of $W_\ell$ necessarily lies in $\operatorname{span}\{I_n, J_n\}$.*

*Remark* C.3. The equivariance condition presented in the theorem is strictly harder than what we need for graph-level, layerwise equivariance:

$$\operatorname{Attn}(h_{\ell-1}(PAP^\top); W_\ell) = P\operatorname{Attn}(h_{\ell-1}(A); W_\ell)\,(I_{K_{\ell-1}} \otimes P^\top).$$

For graphs, it suffices to assume that the hidden states are induced by some $n$-node graph.

*Proof.* STEP 1. RELATING TO WEIGHT CONJUGATION. Fix a layer $\ell$. Write $K = K_{\ell-1}$, $h = h_\ell$, $W = W_\ell$, and let $\sigma(P) = I_K \otimes P$. The first step is to relate the conjugation of hidden states, $T_P(h) : h \mapsto Ph(I_K \otimes P^\top)$, to a conjugation of layer weights, $W_\ell \mapsto \sigma(P)W_\ell\sigma(P)^\top$.

Concretely, since $W_\ell \geq 0$, ReLU. Hence

$$\text{Attn}(T_P(h); W) = \frac{1}{n}\text{ReLU}[(Ph\sigma(P)) \, W \, (\sigma(P)^\top h^\top P^\top)](Ph\sigma(P))$$

$$= \frac{1}{n}P[h\sigma(P) \, W \, (\sigma(P)^\top h^\top)](h\sigma(P))$$

and

$$T_P(\text{Attn}(h; W)) = \frac{1}{n}P(hWh^\top)h\sigma(P).$$

Layer-wise attention equivariance requires the two quantities above to equal for all $h$, and left multiplication by $P^{-1}$ gives

$$h\Delta h^\top h \, \sigma(P) = 0 \qquad \text{for all } h \geq 0 \qquad \text{where} \qquad \Delta := \sigma(P)W\sigma(P)^\top - W. \tag{*}$$

STEP 2. PROVING $\Delta = 0$. To do so, we consider special hidden states, with only two nonzero entries $h_{i,p} = 1$ and $h_{j,q} = t$. Equivalently, pick columns $p \neq q$ and rows/nodes $i \neq k$ and set $h_{i,.} = e_p^\top$, $h_{j,.} = te_q^\top$, and $h = 0$ everywhere else, where $e_p$ is standard basis vector pivoted at $p$.

Because $h$ only uses columns $p$ and $q$, the matrix $h\Delta h^\top$ can be embedded on rows/columns $\{i, j\}$ with values

$$h\Delta h^\top = \begin{pmatrix} \Delta_{p,p} & t\Delta_{p,q} \\ t\Delta_{q,p} & t^2\Delta_{q,q} \end{pmatrix}.$$

Recall $\sigma(P)$ is a permutation on columns; let $\pi$ be the permutation induced by it. Since $h\sigma(P)$ has the same two nonzero rows with $(h\sigma(P))_{i,.} = e_{\pi(p)}^\top$ and $(h\sigma(P))_{j,.} = te_{\pi(q)}^\top$, we get that $(h\Delta h^\top)(h\sigma(P))$ only has rows $i$ and $j$ potentially nonzero:

$$\begin{cases} \text{row } i : \Delta_{p,p}e_{\pi(p)}^\top + t^2\Delta_{p,q}e_{\pi(q)}^\top \\ \text{row } j : t\Delta_{q,p}e_{\pi(p)}^\top + t^2\Delta_{q,q}e_{\pi(q)}^\top. \end{cases}$$

But recall (*): $(h\Delta h^\top)(h\sigma(P)) = 0$ for all $t > 0$. The two standard basis vectors $e_{\pi(p)}, e_{\pi(q)}$ are linearly independent, so the coefficients must be uniformly zero! Hence $\Delta_{p,p} = \Delta_{p,q} = \Delta_{q,p} = \Delta_{q,q} = 0$. Finally, because $p \neq q$ were arbitrary, this forces $\Delta = 0$ entrywise. and that $\sigma(P)W_\ell\sigma(P)^\top = W_\ell$ for this $P$. And because $P$ is arbitrary, we conclude that $\sigma(P)W\sigma(P)^\top = W$ for every permutation $P$.

STEP 3. RELATING TO $n \times n$ BLOCKS. Consider any $n \times n$ block $W[u, v]$ of $W$ where $1 \leq u, v \leq K_\ell$. Using $\sigma(P) = I_{K_\ell} \otimes P^\top$ and taking the $(u, v)$ block on both sides,

$$(\sigma(P)W\sigma(P)^\top)[u, v] = \sum_{a,b}(I_{K_\ell})_{u,a}P^\top W[a, b]P(I_{K_\ell})_{b,v} = P^\top W[u, v]P.$$

The LHS equals $W[u, v]$, so we conclude that

$$P^\top W[u, v]P = W[u, v] \qquad \text{for all } P \in S_n.$$

In other words, layerwise equivariance implies each block must be invariant under $P^\top(\cdot)P$. Taking any transposition forces all diagonal entries of a block to equal, while for any $i \neq j$, $k \neq \ell$, any arbitrary permutation mapping $\pi(i) = k$, $\pi(j) = \ell$ forces entries $(i, j)$ and $(k, \ell)$ to be equal. This implies that each block lies in $\text{span}\{I_n, J_n\}$ as claimed. $\square$

## C.2 Population Gradient Lives in the Equivariant Algebra

**Theorem C.4** (Population gradient lives in the equivariant algebra)**.** *Under Assumption 4.8, in particular using layerwise parameterization $W_\ell = A_\ell \otimes I_n + B_\ell \otimes J_n$, fix a layer $\ell$ and let $K = K_{\ell-1}$. Then the population gradient with respect to $W_\ell$ lies in $M_K(\mathbb{R}) \otimes \text{span}\{I_n, J_n\}$: there exist matrices $G_\ell^{(I)}, G_\ell^{(J)} \in \mathbb{R}^{K \times K}$ such that*

$$\mathbb{E}\Big[\frac{\partial\mathcal{L}}{\partial W_\ell}\Big] = G_\ell^{(I)} \otimes I_n + G_\ell^{(J)} \otimes J_n. \tag{6}$$

*Proof.* We let $S_n$ act on node indices. Since $W_\ell$ can be parametrized as $W_\ell = A_\ell \otimes I_n + B_\ell \otimes J_n$, the attention map is equivariant under left-right action:

$$\text{Attn}(Ph(I_K \otimes P^\top); W_\ell) = P\,\text{Attn}(h; W_\ell)(I_K \otimes P^\top),$$

and so is the full map $A \mapsto Z$. For any fixed permutation $P$, the data $\text{ER}(n, p)$ is permutation-invariant, i.e., $A$ and $PAP^\top$ are identically distributed. Because the model map and the loss are equivariant under $A \mapsto PAP^\top$ with $R \mapsto PRP^\top$, the sample gradient covaries as

$$\nabla_{W_\ell}\mathcal{L}(PAP^\top) = (I_K \otimes P)\nabla_{W_\ell}\mathcal{L}(A)(I_K \otimes P^\top).$$

Taking expectation over $A$ gives

$$\mathbb{E}_A[\nabla_{W_\ell}\mathcal{L}(PAP^\top)] = (I_K \otimes P)\mathbb{E}_A[\nabla_{W_\ell}\mathcal{L}(A)](I_K \otimes P^\top)$$

for every $P$. Hence the population gradient lies in the commutant of $\{I_K \otimes P : P \in S_n\}$. It remains to identify this commutant. View $G_\ell$ as a $K \times K$ block matrix with $n \times n$ sub-blocks. The relation $(I_K \otimes P)^\top G_\ell(I_K \otimes P) = G_\ell$ says each $n \times n$ block $B$ satisfies $P^\top BP$ for all permutations $P$, so the block must have one value on the diagonal and one on the off-diagonals. It is well known that the fixed-point algebra of conjugation on $n \times n$ matrices is $\text{span}(I_n, J_n)$. Hence every block lies in this span, i.e., $G_\ell \in M_K(\mathbb{R}) \otimes \text{span}\{I_n, J_n\}$. $\qquad\square$

## C.3 Which Conditions Encourage $W_\ell \approx A_\ell \otimes I_n$?

To facilitate the following analyses, it will be beneficial to first (re)introduce some notations.

Throughout the analysis of training dynamics, we inherit the notations used in Assumption 4.8: we use $Z$ to denote the model output, $R$ the reachability matrix, $A$ the adjacency matrix, $\mathcal{L} = \mathcal{L}(Z; R)$ the loss, and $\mathcal{R}(\Theta)$ the population risk $\mathcal{R}(\Theta) := \mathbb{E}_{G \sim \text{ER}(n,p)}[\mathcal{L}(\text{TF}_\Theta^L(A_G); R_G)]$.

Fix a layer $\ell$ and a nonnegative direction $\Delta \geq 0$ in the $J$-channel. Write $D = \frac{\partial Z}{\partial B_\ell}[\Delta]$ (more details in Theorem C.5). We say a node pair $(i, j)$ is **active** for $\Delta$ if $D_{i,j} > 0$. In particular, we say $\Delta$ is active on cross-component pairs if $D_{i,j} > 0$ for some $(i, j)$ belonging to different connected components (note $\Delta$ could also be active on within-component pairs).

Because we constrain $W_\ell \geq 0$, under the parameterization $W_\ell = A_\ell \otimes I_n + B_\ell \otimes J_n$, we must also have $B_\ell \geq 0$. Then, the appropriate notion of stationarity is KKT: in our setting, this reduces to

$$\nabla_{B_\ell}\mathcal{R}(\Theta) \geq 0, \qquad B_\ell \geq 0, \qquad \text{and} \qquad \nabla_{B_\ell}\mathcal{R}(\Theta) \odot B_\ell = 0$$

which we use in the Theorem below.

**Theorem C.5** (Population Training Conditionally Suppresses the $J$-Channel). *Assume Assumption 4.8. Fix any layer $\ell$ and decompose $W_\ell = A_\ell \otimes I_n + B_\ell \otimes J_n$. Let $Z$ be the output, $R$ the reachability matrix (ground truth), $\mathcal{L} = \mathcal{L}(Z; R)$ the loss, and $\mathcal{R}(\Theta)$ the population risk.*

1. (***Directional derivative on nonnegative $J$-channel directions.***) *Let $\Delta \in \mathbb{R}^{K_{\ell-1} \times K_{\ell-1}}$ be entrywise nonnegative and define the one-sided Fréchet derivative $D := \frac{\partial Z}{\partial B_\ell}[\Delta] \in \mathbb{R}^{n \times n}$. Then $D \geq 0$ entrywise, and the population directional derivative satisfies*

$$D_{B_\ell}\mathcal{R}(\Theta)[\Delta] = \mathbb{E}\left\langle \left[\frac{\partial \mathcal{L}}{\partial Z}, D\right]\right\rangle_F = \alpha \cdot \mathbb{E}\left[\underbrace{\sum_{R_{i,j}=0} D_{i,j}}_{\substack{\text{cross component} \\ \text{penalty}}} - \underbrace{\sum_{R_{i,j}=1} \frac{1 - \phi_\epsilon(Z_{i,j})}{\phi_\epsilon(Z_{i,j})} D_{i,j}}_{\substack{\text{within-component} \\ \text{reward}}}\right]. \tag{7}$$

   *In particular, $D_{B_\ell}\mathcal{R}(\Theta)[\Delta] \geq 0$ iff the Population-Level Dominance Condition holds (i.e. Equation (7) is positive). Throughout this Appendix, we will use "cross component penalty" and "within-component reward" to denote these two competing terms.*

2. (***Consequences for KKT stationary points.***) *Assume $\Theta$ is KKT-stationary for $B_\ell \geq 0$:*

$$\nabla_{B_\ell}\mathcal{R}(\Theta) \geq 0, \qquad B_\ell \geq 0, \qquad \text{and} \qquad \nabla_{B_\ell}\mathcal{R}(\Theta) \odot B_\ell = 0 \tag{8}$$

*Let $\Delta = |B_\ell|$ (entrywise absolute value) and let $D = \frac{\partial Z}{\partial B_\ell}[|B_\ell|]$. If, with positive probability under $\mathsf{ER}(n,p)$, $\Delta$ activates at least one cross-component pair, and if the Population-Level Dominance Condition holds, then $B_\ell = 0$. Equivalently, under activation at $\Delta = |B_\ell|$ and strict dominance by cross-component penalty, the only KKT stationary point in the $J_n$-channel is $B_\ell = 0$.*

**Lemma C.6** (Monotonicity in the $J$-channel). *Fix $\ell$ and hold all parameters except $B_\ell$. Write $h_{\ell-1} = [X_1 \mid \dots \mid X_{K_{\ell-1}}]$ and $u_p = X_p \mathbf{1} \in \mathbb{R}^n_{\geq 0}$. Then*

$$h_{\ell-1} W_\ell h_{\ell-1}^\top = \sum_{p,q} (A_\ell)_{p,q} X_p X_q^\top + \sum_{p,q} (B_\ell)_{p,q} u_p u_q^\top. \tag{9}$$

*Consequently, for every nonnegative direction $\Delta \geq 0$ in the $J$-channel, the one-sided Fréchet derivative at $0^+$ exists and is entrywise nonnegative. Hence, along the ray $\{B_\ell + \delta\Delta \mid \delta \geq 0\}$, the output is entrywise nondecreasing:*

$$\frac{\partial Z}{\partial B_\ell}[\Delta] \in \mathbb{R}^{n \times n}_{\geq 0}, \qquad Z(B_\ell + \delta\Delta) - Z(B_\ell) \geq 0 \text{ for all } \delta \geq 0.$$

*Moreover, if $G$ is disconnected, and either (i) $\Delta_{p,p} > 0$ for a block $p$ such that $u_p$ has support in at least two components, or (ii) there exist blocks $p, q$ with $\Delta_{p,q} > 0$ and $u_p, u_q$ supported in different components, then there exist cross component pairs $(i,j)$ with $(\frac{\partial Z}{\partial B_\ell}[\Delta])_{i,j} > 0$.*

*Proof.* Since $J_n x = (\mathbf{1}^\top x)\mathbf{1}$ for $x \in \mathbb{R}^n$, we have $X_p J_n X_q^\top = (X_p \mathbf{1})(X_q \mathbf{1})^\top = u_p u_q^\top$, yielding the displayed decomposition. For $B_\ell \mapsto B_\ell + \delta\Delta$ with $\Delta \geq 0$, the layer scores

$$R_\ell(B_\ell + \delta\Delta) - R_\ell(B_\ell) = \delta \sum_{p,q} \Delta_{p,q} u_p u_q^\top \geq 0,$$

so the one-sided derivative exists and is entrywise nonnegative. Because all subsequent maps are entrywise monotone, this implies $Z(B_\ell + \delta\Delta) - Z(B_\ell) \geq 0$ as stated.

For the "moreover" part, in case (i), $u_p u_p^\top$ places positive mass on index pairs spanning the components where $u_p > 0$, and in case (ii), $u_p u_q^\top$ (or its transpose) places positive mass across two components supporting $u_p$ and $u_q$. Monotonicity propagates these positives to $D = \frac{\partial Z}{\partial B_\ell}[\Delta]$. $\qquad\square$

*Proof of Theorem C.5.* For the population risk $\mathcal{R}(\Theta) = \mathbb{E}[\mathcal{L}(Z; R)]$, applying definitions gives the directional derivative along $\Delta$ gives

$$D_{B_\ell}\mathcal{R}(\Theta)[\Delta] = \left\langle \mathbb{E}\left[\frac{\partial \mathcal{L}}{\partial Z}\right], D \right\rangle_F = \alpha \cdot \mathbb{E}\left[\sum_{i,j}\left(1 - \frac{R_{i,j}}{\phi_\epsilon(Z_{i,j})}\right) D_{i,j}\right].$$

Separating indices by $R_{i,j} \in \{0,1\}$ proves equation 7.

For the second claim, evaluate equation 7 at $\Delta = |B_\ell|$. Under the activation premise (Lemma C.6) and strict dominance by cross-component penalty, we obtain $D_{B_\ell}\mathcal{R}(\Theta)[|B_\ell|] = \langle \nabla_{B_\ell}\mathcal{R}(\Theta), \Delta \rangle_F > 0$. Since $\nabla_{B_\ell}\mathcal{R}(\Theta) \geq 0$ and $|B_\ell| \geq 0$, a strictly positive inner product violates the KKT complementary condition $\nabla_{B_\ell}\mathcal{R}(\Theta) \odot B_\ell = 0$ unless $B_\ell = 0$. $\qquad\square$

*Remark* C.7. While Theorem C.5 mostly discusses the suppression of $B_\ell$, its (i) in fact reveals a quite interesting, opposite phenomenon: **early training promotes** $B_\ell$. Before the model learns to pick up easy connected pairs, the corresponding values $\phi_\epsilon(Z_{i,j}) \ll 1$. Consequently, the fractions $(1 - \phi_\epsilon(Z_{i,j}))/\phi_\epsilon(Z_{i,j})$ are large, making equation 7 negative. Gradient descent then pushes $B_\ell$ up "without feeling pressure." As training proceeds, these easy connected pairs saturate ($\phi_\epsilon(Z_{i,j}) \to 1$), while simultaneously $\Delta$ begins to active cross pairs (the "moreover" part of Lemma C.6), increasing the $R = 0$ term in equation 7 and potentially flipping the sign. This is when the $J$-channel starts to incur penalty. This explains the transient "Phase 1" in §4.3.

*Remark* C.8. The $B_\ell \otimes J_n$-channel injects rank-one dense terms $u_p u_q^\top$ into the attention core. On disconnected graphs, these terms produce cross-component positives, which the reachability target $R$ labels as negatives. Because disconnected graphs appear with positive probability in the data, the population gradient penalizes every nonnegative direction in the $J$-channel active on cross-component pairs whenever the cross-component penalty dominates within-component reward. Under the same activation and cross-component penalty dominance assumptions, any KKT stationary point must have $B_\ell = 0$. In short: under these conditions, population drives the node-side factor towards locality, i.e., $W_\ell \approx A_\ell \otimes I_n$.

## C.4 Which Samples Push Which Channel? (Local $I_n$ vs. Global $J_n$)

Recall $W_\ell = A_\ell \otimes I_n + B_\ell \otimes J_n$ and Lemma C.6. The $I$-channel controls local propagation within components; the $J$-channel couples to the global / mean direction and injects dense rank-one terms. In this section, we first shift to a micro-level perspective, focusing on the effects of individual samples (graphs), and then draw connection to how the training distribution determines the model's eventual behavior (algorithmic vs. heuristic, §4.3).

We decompose the single-sample loss $\mathcal{L}_G(\Theta) := \mathcal{L}(\mathsf{TF}_\Theta^L(A); R)$ and examine directional derivatives at a fixed $\Theta$, with the link gradient $\partial\mathcal{L}/\partial Z = \alpha(1 - R/\phi_\epsilon(Z))$. Throughout, we say a pair $(i, j)$ is **saturated** if its per-pair loss gradient vanishes; for within-component pairs ($R_{i,j} = 1$) this is equivalent to $\phi_\epsilon(Z_{i,j}) = R_{i,j}$. We say a direction $\Delta$ is **active** over $(i, j)$ if the corresponding channel directive $D_{i,j} > 0$, where $D$ denotes $\frac{\partial Z}{\partial A_\ell}[\Delta]$ or $\frac{\partial Z}{\partial B_\ell}[\Delta]$ as appropriate.

Our first main result is the following Theorem, which intuitively claims two things:

- *(Within capacity) Small-diameter graphs "reward" the local $I$-channel and, if disconnected, penalizes the global $J$-channel if activated.*

- *(Beyond capacity) Large-diameter connected graphs demand a global shortcut: the $J$-channel is promoted, while the $I$-channel remains confined to short-range corrections.*

**Theorem C.9** (Per-sample pushes by diameter). *Fix a layer $\ell$ and nonnegative directions $\Delta_A, \Delta_B \geq 0$ for $A_\ell, B_\ell$, respectively. Assume $B_1 = \ldots = B_L = 0$.*

(i) *(Within capacity) If $\mathrm{diam}(G) \leq 3^L$, then $D_{A_\ell}\mathcal{L}_G(\Theta)[\Delta_A] \leq 0$, with strict $< 0$ whenever $\Delta_A$ is active on at least one unsaturated within-component pair. If, in addition, $G$ is disconnected, then $D_{B_\ell}\mathcal{L}_G(\Theta)[\Delta_B] > 0$ if both of the following hold: $\Delta_B$ is active at at least one cross-component pair, and Population-Level Dominance Condition holds.*

(ii) *(Beyond capacity) If $\mathrm{diam}(G) > 3^L$ and $G$ is connected, then we have $D_{A_\ell}\mathcal{L}_G(\Theta)[\Delta_A] \leq 0$ where only within-capacity pairs can contribute, and $D_{B_\ell}\mathcal{L}_G(\Theta)[\Delta_B] < 0$ for $\Delta_B$ that is active on at least one unsaturated pair.*

To prove this Theorem, we split the argument into the following four lemmas, each isolating one ingredient of the dynamics. Firstly, Lemma C.10 shows that the local $I$-channel is monotone: any nonnegative $A_\ell$ cannot increase the loss and is strictly helpful on unsaturated within-component pairs. This lets us treat local corrections as "harmless," while Lemma C.11 analyze the sign of the global $J$-channel (connected vs. disconnected), and Lemma C.12 determines which pairs are ever affected when $B = 0$. Together, they yield the two cases in Theorem C.9.

**Lemma C.10** (Local channel always helps). *Assume $B_1 = \ldots = B_L = 0$. For any graph $G$, any layer $\ell$, and any direction $\Delta \geq 0$ in the $I$-channel,*

$$D_{A_\ell}\mathcal{L}_G(\Theta)[\Delta] = \left\langle \frac{\partial\mathcal{L}}{\partial Z}, \frac{\partial Z}{\partial A_\ell}[\Delta] \right\rangle_F \leq 0, \tag{10}$$

*with strict inequality whenever there exists a within-component, unsaturated pair, on which $\Delta$ is active.*

*Proof.* From the block decomposition from equation 9, the $I$-channel contributes $\sum_{p,q} \Delta_{p,q} X_p X_q^\top$., which is block-diagonal with respect to the component partition. Hence $\frac{\partial Z}{\partial A_\ell}[\Delta]$ has support only on pairs $(i, j)$ in the same component. On those pairs, $R_{i,j} = 1$, and thus

$$\left(\frac{\partial\mathcal{L}}{\partial Z}\right)_{i,j} = \alpha \cdot \left(1 - \frac{1}{\phi_\epsilon(Z_{i,j})}\right) = -\alpha \cdot \frac{1 - \phi_\epsilon(Z_{i,j})}{\phi_\epsilon(Z_{i,j})} \leq 0,$$

with strict negativity whenever $\phi_\epsilon(Z_{i,j}) < 1$. Entrywise, nonnegativity of the forward map (Lemma C.6) gives $\frac{\partial Z}{\partial A_\ell}[\Delta] \geq 0$. Therefore the Frobenius inner product $\leq 0$, and $< 0$ under the stated conditions. $\square$

We now switch from the local $I$-channel to the global $J$-channel and will use that the forward sensitivity in the $J$-channel is entrywise nonnegative, so the sign of the directional derivative is controlled entirely by the per-pair loss gradient.

**Lemma C.11** (Global channel helps connected graphs and conditionally hurts disconnected graphs). *Fix a layer $\ell$ and a nonnegative direction $\Delta \geq 0$ in the $J$-channel.*

*(i)* If $G$ is connected, then $D_{B_\ell}\mathcal{L}_G(\Theta) \leq 0$, with strict $< 0$ whenever there exists an unsaturated pair $(i,j)$ (i.e., $\phi_\epsilon(Z_{i,j}) < 1$) on which $\Delta$ is active ($D_{i,j} > 0$).

*(ii)* If $G$ is disconnected, then

$$D_{B_\ell}\mathcal{L}_G(\Theta)[\Delta] = \alpha \cdot \left[ \sum_{R_{i,j}=0} D_{i,j} - \sum_{R_{i,j}=1} \frac{1 - \phi_\epsilon(Z_{i,j})}{\phi_\epsilon(Z_{i,j})} D_{i,j} \right], \tag{11}$$

hence $D_{B_\ell}\mathcal{L}_G(\Theta)[\Delta] \geq 0$ whenever the Population-Level Dominance Condition holds. Strict $> 0$ holds if the inequality is strict, and $\Delta$ is active on at least one cross pair.

*Proof.* By the chain rule,

$$D_{B_\ell}\mathcal{L}_G(\Theta)[\Delta] = \left\langle \frac{\partial\mathcal{L}}{\partial Z}, \frac{\partial Z}{\partial B_\ell}[\Delta] \right\rangle_F = \left\langle \alpha \cdot \left( 1 - \frac{R}{\phi_\epsilon(Z)} \right), D \right\rangle_F. \tag{12}$$

By Lemma C.6 , $D \geq 0$ entrywise; moreover, $D_{i,j} > 0$ exactly on pairs where $\Delta$ is active.

*(i)* If $G$ is connected, then the reachability matrix $R$ is all-ones. Hence $\frac{\partial\mathcal{L}}{\partial Z} = -\alpha(1 - \phi_\epsilon(Z))/\phi_\epsilon(Z) \leq 0$ entrywise, with strict negativity whenever $\phi_\epsilon(Z_{i,j}) < 1$. Pairing with $D \geq 0$ and $D_{i,j} > 0$ on active pairs gives $D_{B_\ell}\mathcal{L}_G(\Theta)[\Delta] \leq 0$ and strict $< 0$ under the stated saturation / activation conditions.

*(ii)* If $G$ is disconnected, split equation 12 over $R_{i,j} = 0$ and $R_{i,j} = 1$ to obtain the displayed identity. Since $D \geq 0$, the stated dominance condition yields $\geq 0$. Strictness requires a cross pair with $D_{i,j} > 0$, holds exactly when $\Delta$ is active on at least one cross-component pair. □

We now show that when the global $J$-channel is disabled, the model can only light up within-capacity pairs. Note this is somewhat a converse to Theorem C.2, where a "good" model that only lights up within-capacity pairs necessarily have each $W_\ell[r,s]$ diagonal. The following Lemma isolates the role of the $J$-channel as the only "nontrivial" shortcut.

Recall from Definition 4.6: for a depth $L$ and a graph $G$ with adjacency matrix $A$, we call a pair $(i,j)$ **within capacity** if $[A^{3^L}]_{i,j} > 0$ and **beyond capacity** otherwise.

**Lemma C.12** (*I*-channel reaches within-capacity pairs; *J*-channel is the only dense shortcut). *At any $\Theta$ with $B_1 = \ldots = B_L = 0$, the output satisfies*

$$Z_{i,j} > 0 \implies [A^{3^L}]_{i,j} > 0.$$

*Equivalently, beyond-capacity pairs receive no positive mass from the I-channel alone. In contrast, for any $\ell$ and any $\Delta \geq 0$ in the J-channel, $\frac{\partial Z}{\partial B_\ell}[\Delta] \geq 0$ and is strictly positive on active pairs by definition.*

*Proof.* Since $B_\ell = 0$ implies $h_{\ell-1}W_\ell h_{\ell-1}^\top = \sum_{p,q}(A_\ell)_{p,q}X_pX_q^\top$ from Lemma C.6, it is easy to see that they are block-diagonal w.r.t. connected components. Hence Lemma B.2 applies and support expands by at most a factor of 3 per layer, and only within-capacity pairs receive mass. The density statement follows from Lemma C.6: For any $\Delta \geq 0$ in the $J$-channel, we have $\frac{\partial Z}{\partial B_\ell} = \sum_{p,q} \Delta_{p,q} u_p u_q^\top u \geq 0$. The strict positiveness characterization follows directly from Lemma C.6. □

With the previous lemmas established, we can now assemble the per-sample sign rules. Intuitively, the $I$-channel makes only local corrections, never hurting the loss and only touching within-capacity pairs when $B = 0$, while the $J$-channel is the sole dense shortcut, helpful on connected graphs but penalized by cross-component pairs when the graph is disconnected.

*Proof of Theorem C.9.* Let $\ell, \Delta_A, \Delta_B$ be given as described. Set $D_A = \frac{\partial Z}{\partial A_\ell}[\Delta_A]$ and $D_B = \frac{\partial Z}{\partial B_\ell}[\Delta_B]$. Recall from chain rule

$$D_{(\cdot)}\mathcal{L}_G(\Theta)[\cdot] = \left\langle \frac{\partial\mathcal{L}}{\partial Z}, \frac{\partial Z}{\partial(\cdot)}[\cdot] \right\rangle_F = \alpha \cdot \left\langle 1 - \frac{R}{\phi_\epsilon(Z)}, \frac{\partial Z}{\partial(\cdot)}[\cdot] \right\rangle_F.$$

*(i)* (Within capacity) By Lemma C.10, for any $\Delta_A \geq 0$ the $I$-channel directional derivative is $\leq 0$, with strict inequality under the stated conditions. The result on disconnected graphs $G$ follows from Lemma C.11.

(ii) By Lemma C.12, with $B = 0$, only within-capacity pairs can be affected by the $I$-channel, so Lemma C.10 gives $D_{A_\ell}\mathcal{L}_G(\Theta)[\Delta_A] \leq 0$. Since $G$ is connected and $\mathrm{diam}(G) > 3^L$, there will be unsaturated pairs; then Lemma C.11(i) yields $D_{B_\ell}\mathcal{L}_G(\Theta)[\Delta_B] < 0$, as claimed. $\qquad\square$

*Remark* C.13 (Population-level consequence under $\mathrm{ER}(n,p)$). Fix a layer $\ell$ and nonnegative directions $\Delta_A, \Delta_B \geq 0$. Partition the graphs into $\mathcal{G}_0 = \{G : \mathrm{diam}(G) \leq 3^L\}$ and $\mathcal{G}_1 = \{G : \mathrm{diam}(G) > 3^L\}$. Writing the population directional derivatives as mixtures,

$$D_{B_\ell}\mathcal{R}(\Theta)[\Delta_B] = \mathbb{P}(\mathcal{G}_0)\mathbb{E}[D_{B_\ell}\mathcal{L}_G(\Theta)[\Delta_B] \mid G \in \mathcal{G}_0] + \mathbb{P}(\mathcal{G}_1)\mathbb{E}[D_{B_\ell}\mathcal{L}_G(\Theta)[\Delta_B] \mid G \in \mathcal{G}_1]. \tag{13}$$

We claim the following on the population gradient.

(i) (Local) From Lemma C.10, once the global $J$-channel has been suppressed, the local $I$-channel is consistently promoted until saturation.

(ii) (Global) The population gradient along the global $J$-channel is an explicit mixture of two regimes: large, connected graphs beyond capacity that promote the $J$-channel, and small, disconnected graphs within capacity that suppress it whenever cross-component errors persist. Formally:

(ii.a) If $G$ is connected and $\mathrm{diam}(G) > 3^L$, then by Lemma C.12, every beyond-capacity pair has $Z_{ij} = 0$ while $R_{ij} = 1$. For those pairs, we have $\partial\mathcal{L}/\partial Z = -\alpha(1 - \phi_\epsilon(Z))/\phi_\epsilon(Z) < 0$. By Lemma C.6, the inner product $\langle\partial\mathcal{L}/\partial Z, \partial Z/\partial B_\ell[\Delta_B]\rangle_F < 0$ too. Integrating over all beyond-capacity, connected graphs yields

$$\mathbb{E}[D_{B_\ell}\mathcal{L}_G(\Theta)[\Delta_B] \mid G \in \mathcal{G}_1 \ \& \ G \text{ connected}] < 0. \tag{14}$$

(ii.b) If $G$ is disconnected and $\mathrm{diam}(G) \leq 3^L$, then by Lemma C.11, $D_{B_\ell}\mathcal{L}_G(\Theta)[\Delta_B] \geq 0$ with strict $> 0$ if cross-component errors persist (the $\sum_{R=0} D$ term strictly dominates the $\sum_{R=1}(1 - \phi_\epsilon(Z))/\phi_\epsilon(Z) \cdot D$ term), and if $\Delta_B$ is active on cross pairs (i.e. $D_{ij}^{(B)} > 0$ for some $R_{ij} = 0$). The latter holds by Lemma C.6 if $\Delta_B$ is active on at least one cross pair. Integrating thus yields

$$\mathbb{E}\big[D_{B_\ell}\mathcal{L}_G(\Theta)[\Delta_B] \,\big|\, G \in \mathcal{G}_0 \ \& \ G \text{ disconnected}\big] \geq 0, \tag{15}$$

and strictly $> 0$ provided the two additional assumptions above.

## C.5 Convergence of Projected Gradient Descent to KKT Points

In this subsection, we establish that projected gradient descent on the regularized population risk converges to points satisfying the Karush-Kuhn-Tucker (KKT) conditions. This justifies the stationarity assumption underlying Theorem C.5. We begin by stating the main result.

**Theorem 4.9.** *Let $\mathcal{R}(\Theta) := \mathbb{E}_{G\sim\mathrm{ER}(n,p)}[\mathcal{L}(\mathsf{TF}_\Theta^L(A_G); R_G)]$ denote the population risk. For $\lambda > 0$, define the regularized objective $\mathcal{R}_\lambda(\Theta) := \mathcal{R}(\Theta) + \frac{\lambda}{2}\|\Theta\|_F^2$. Let $\mathcal{C} := \{(A_\ell, B_\ell)_\ell : A_\ell \geq 0, B_\ell \geq 0, \forall\ell\}$ denote the constraint set, and consider the sequence $\{\Theta^{(k)}\}_{k\geq 0}$ generated by projected gradient descent on $\mathcal{R}_\lambda$:*

$$\Theta^{(k+1)} = \Pi_\mathcal{C}\left(\Theta^{(k)} - \eta\nabla\mathcal{R}_\lambda(\Theta^{(k)})\right), \tag{16}$$

*with step size $\eta > 0$ sufficiently small and initialization $\Theta^{(0)} \in \mathcal{C}$ of the form $W_\ell = A_\ell \otimes I + B_\ell \otimes J$. Then every limit point $\Theta_\lambda^* \in \mathcal{C}$ satisfies the KKT conditions:*

$$\nabla_{B_\ell}\mathcal{R}(\Theta_\lambda^*) + \lambda B_\ell^* \geq 0, \quad B_\ell^* \geq 0, \quad (\nabla_{B_\ell}\mathcal{R}(\Theta_\lambda^*) + \lambda B_\ell^*) \odot B_\ell^* = 0, \tag{17}$$

*and analogously for $A_\ell^*$. Moreover, after $K$ iterations, there exists $k < K$ such that $\Theta^{(k)}$ is an $\epsilon$-approximate KKT point with $\epsilon = O(1/K)$.*

The proof adapts the standard convergence analysis for gradient descent on smooth nonconvex functions (Nesterov, 2004) to the projected setting. The argument proceeds in three stages: we first introduce the gradient mapping as the appropriate measure of stationarity for constrained problems, then establish a sufficient decrease property for each iteration, and finally combine these ingredients via a telescoping argument to obtain the convergence rate.

### C.5.1 PRELIMINARIES

We work in the decomposed parameter space $\Theta = (A_\ell, B_\ell)_\ell$, where $W_\ell = A_\ell \otimes I + B_\ell \otimes J$ as in Assumption 4.8. By Theorem C.4, if the initialization lies in this subalgebra, then the gradient $\nabla \mathcal{R}_\lambda(\Theta)$ also decomposes as a direct sum over the $(A_\ell, B_\ell)$ components, and projected gradient descent preserves this structure. The constraint set $\mathcal{C} = \{(A_\ell, B_\ell)_\ell : A_\ell \geq 0, B_\ell \geq 0\}$ is the non-negative orthant in this decomposition, and the projection $\Pi_\mathcal{C}$ acts component-wise as $\Pi_\mathcal{C}(\Theta) = \max\{0, \Theta\}$.

Recall that a point $\Theta^* \in \mathcal{C}$ satisfies the KKT conditions for minimizing $\mathcal{R}_\lambda$ over $\mathcal{C}$ if

$$\nabla \mathcal{R}_\lambda(\Theta^*) \geq 0, \quad \Theta^* \geq 0, \quad \text{and} \quad \nabla \mathcal{R}_\lambda(\Theta^*) \odot \Theta^* = 0, \tag{18}$$

where $\odot$ denotes the Hadamard product and inequalities hold component-wise. These conditions assert that interior components (where $\Theta^* > 0$) have vanishing gradient, while boundary components (where $\Theta^* = 0$) have non-negative gradient pointing outward from the feasible region.

In unconstrained optimization, the gradient norm $\|\nabla f(\Theta)\|$ measures proximity to stationarity. For constrained problems, the appropriate generalization is the gradient mapping.

**Definition C.14** (Gradient Mapping). For step size $\eta > 0$, the gradient mapping at $\Theta \in \mathcal{C}$ is

$$G_\eta(\Theta) := \frac{1}{\eta} \left(\Theta - \Pi_\mathcal{C}\left(\Theta - \eta \nabla \mathcal{R}_\lambda(\Theta)\right)\right). \tag{19}$$

The gradient mapping quantifies the displacement induced by a projected gradient step: the update rule can be written as $\Theta^{(k+1)} = \Theta^{(k)} - \eta G_\eta(\Theta^{(k)})$. When $\mathcal{C}$ is unconstrained, the projection is the identity and $G_\eta(\Theta) = \nabla \mathcal{R}_\lambda(\Theta)$. The following lemma confirms that the gradient mapping vanishes precisely at KKT points.

**Lemma C.15.** *For any $\eta > 0$, a point $\Theta^* \in \mathcal{C}$ satisfies $G_\eta(\Theta^*) = 0$ if and only if $\Theta^*$ is a KKT point.*

*Proof.* The condition $G_\eta(\Theta^*) = 0$ is equivalent to $\Theta^* = \Pi_\mathcal{C}(\Theta^* - \eta \nabla \mathcal{R}_\lambda(\Theta^*))$. For the non-negative orthant, this becomes

$$\Theta^* = \max\left\{0, \Theta^* - \eta \nabla \mathcal{R}_\lambda(\Theta^*)\right\}. \tag{20}$$

On the interior $\{\Theta^* > 0\}$, equality requires $\nabla \mathcal{R}_\lambda(\Theta^*) = 0$. On the boundary $\{\Theta^* = 0\}$, the condition reduces to $0 = \max\{0, -\eta \nabla \mathcal{R}_\lambda(\Theta^*)\}$, which holds if and only if $\nabla \mathcal{R}_\lambda(\Theta^*) \geq 0$. These are precisely the KKT conditions. $\square$

### C.5.2 REGULARITY OF THE OBJECTIVE

The convergence analysis requires two properties of the regularized objective: coercivity, which ensures that iterates remain bounded, and smoothness, which enables a descent inequality.

**Lemma C.16** (Coercivity). *For any $\lambda > 0$, the sublevel sets of $\mathcal{R}_\lambda$ are bounded: if $\mathcal{R}_\lambda(\Theta) \leq c$, then $\|\Theta\|_F \leq \sqrt{2c/\lambda}$.*

*Proof.* Since $\mathcal{R}(\Theta) \geq 0$, we have $\mathcal{R}_\lambda(\Theta) \geq \frac{\lambda}{2}\|\Theta\|_F^2$. The bound follows by rearrangement. $\square$

Coercivity is essential: without regularization, the scaling symmetry of ReLU networks ($W_\ell \mapsto \alpha W_\ell$, $W_{\ell+1} \mapsto \alpha^{-1} W_{\ell+1}$) renders the sublevel sets unbounded, and iterates could escape to infinity.

**Lemma C.17** (Lipschitz Gradient). *Under Assumption 4.8, for any bounded set $\mathcal{B} \subseteq \mathcal{C}$, there exists $L_\mathcal{B} > 0$ such that*

$$\|\nabla \mathcal{R}_\lambda(\Theta_1) - \nabla \mathcal{R}_\lambda(\Theta_2)\| \leq L_\mathcal{B}\|\Theta_1 - \Theta_2\| \quad \textit{for all } \Theta_1, \Theta_2 \in \mathcal{B}. \tag{21}$$

*In particular, $\nabla \mathcal{R}_\lambda$ is Lipschitz continuous on the sublevel set $\{\Theta \in \mathcal{C} : \mathcal{R}_\lambda(\Theta) \leq \mathcal{R}_\lambda(\Theta^{(0)})\}$.*

*Proof.* Within the constraint set $\mathcal{C}$, ReLU activations act as the identity. Since the adjacency matrix $A_G \geq 0$ and $W_\ell = A_\ell \otimes I + B_\ell \otimes J \geq 0$ for $(A_\ell, B_\ell) \in \mathcal{C}$, induction shows $h_{\ell-1} W_\ell h_{\ell-1}^\top \geq 0$ at every layer, so ReLU is the identity throughout the forward pass. The transformer output is therefore a polynomial in $\Theta$, the loss function is smooth (with $\phi_\epsilon \geq \epsilon > 0$ keeping the logarithm away from zero), and the regularization is quadratic. The composition is smooth on $\mathcal{C}$, and smooth functions have Lipschitz gradients on bounded sets. The second statement follows from Lemma C.16, which ensures the sublevel set is bounded. $\square$

### C.5.3 SUFFICIENT DECREASE

The core of the analysis is a progress bound showing that each projected gradient step decreases the objective by an amount proportional to the squared gradient mapping norm.

**Lemma C.18** (Sufficient Decrease). *For step size $\eta \leq 1/L$, the iterates satisfy*

$$\mathcal{R}_\lambda(\Theta^{(k+1)}) \leq \mathcal{R}_\lambda(\Theta^{(k)}) - \frac{\eta}{2}\|G_\eta(\Theta^{(k)})\|^2. \tag{22}$$

*Proof.* Let $\Theta = \Theta^{(k)}$, $\Theta^+ = \Theta^{(k+1)}$, $g = \nabla\mathcal{R}_\lambda(\Theta)$, and $G = G_\eta(\Theta)$. The descent lemma for $L$-smooth functions gives

$$\mathcal{R}_\lambda(\Theta^+) \leq \mathcal{R}_\lambda(\Theta) + \langle g, \Theta^+ - \Theta\rangle + \frac{L}{2}\|\Theta^+ - \Theta\|^2. \tag{23}$$

Since $\Theta^+ - \Theta = -\eta G$, this becomes

$$\mathcal{R}_\lambda(\Theta^+) \leq \mathcal{R}_\lambda(\Theta) - \eta\langle g, G\rangle + \frac{L\eta^2}{2}\|G\|^2. \tag{24}$$

It remains to show that $\langle g, G\rangle \geq \|G\|^2$. The projection $\Theta^+ = \Pi_\mathcal{C}(\Theta - \eta g)$ satisfies the first-order optimality condition: for all $z \in \mathcal{C}$,

$$\langle (\Theta - \eta g) - \Theta^+, z - \Theta^+\rangle \leq 0. \tag{25}$$

Taking $z = \Theta \in \mathcal{C}$ and using $\Theta - \Theta^+ = \eta G$ yields $\langle \eta G - \eta g, \eta G\rangle \leq 0$, which simplifies to $\langle g, G\rangle \geq \|G\|^2$. Substituting into equation 24 and using $\eta \leq 1/L$ completes the proof. $\square$

### C.5.4 PROOF OF THEOREM 4.9

*Proof.* We establish each component of the theorem in turn.

**Bounded iterates.** The sufficient decrease property (Lemma C.18) implies that $\mathcal{R}_\lambda(\Theta^{(k)})$ is non-increasing, so all iterates lie in the initial sublevel set. By Lemma C.16, this set is bounded: $\|\Theta^{(k)}\|_F \leq \sqrt{2\mathcal{R}_\lambda(\Theta^{(0)})/\lambda}$ for all $k$.

**Non-asymptotic rate.** Rearranging Lemma C.18 and summing from $k = 0$ to $K - 1$:

$$\sum_{k=0}^{K-1}\|G_\eta(\Theta^{(k)})\|^2 \leq \frac{2}{\eta}\sum_{k=0}^{K-1}\left[\mathcal{R}_\lambda(\Theta^{(k)}) - \mathcal{R}_\lambda(\Theta^{(k+1)})\right] = \frac{2}{\eta}\left[\mathcal{R}_\lambda(\Theta^{(0)}) - \mathcal{R}_\lambda(\Theta^{(K)})\right]. \tag{26}$$

The right-hand side telescopes. Since $\mathcal{R}_\lambda \geq 0$, we obtain

$$\sum_{k=0}^{K-1}\|G_\eta(\Theta^{(k)})\|^2 \leq \frac{2\mathcal{R}_\lambda(\Theta^{(0)})}{\eta}. \tag{27}$$

The minimum of $K$ non-negative terms is at most their average:

$$\min_{k=0,\ldots,K-1}\|G_\eta(\Theta^{(k)})\|^2 \leq \frac{2\mathcal{R}_\lambda(\Theta^{(0)})}{\eta K}. \tag{28}$$

Thus, within $K$ iterations, at least one iterate achieves $\|G_\eta(\Theta^{(k)})\|^2 \leq \epsilon$ for $\epsilon = O(1/K)$.

**Asymptotic convergence.** Taking $K \to \infty$, the bound $\sum_{k=0}^{\infty}\|G_\eta(\Theta^{(k)})\|^2 < \infty$ implies $\|G_\eta(\Theta^{(k)})\| \to 0$.

**Limit points are KKT.** Boundedness of the iterates follows from Lemma C.16. That every limit point of projected gradient descent on a smooth function over a closed convex set satisfies the first-order stationarity condition is a standard result; see, e.g., Bertsekas (1997) (Proposition 2.3.2) or Beck (2017) (Theorem 10.15). The required smoothness holds by Lemma C.17. Restricting to the $B_\ell$ components and expanding $\nabla\mathcal{R}_\lambda = \nabla\mathcal{R} + \lambda\Theta$ yields equation 17. $\square$

*Remark* C.19 (Sufficiency for Main Results). **Theorem** 4.9 establishes convergence to KKT points in the sense of limit points; it does not rule out the possibility that the sequence oscillates between multiple KKT points. Establishing convergence to a unique limit requires additional structure, such as the Kurdyka-Łojasiewicz property (Attouch et al., 2013). However, for our purposes, convergence to limit points suffices: Theorem C.5 shows that *any* KKT point satisfying our assumptions has $B_\ell^* = 0$, so the heuristic channel is suppressed regardless of which limit point is approached.

C.5.5 CONNECTING REGULARIZED CONVERGENCE TO HEURISTIC SUPPRESSION

Theorem 4.9 establishes that projected gradient descent on the regularized objective $\mathcal{R}_\lambda(\Theta) = \mathcal{R}(\Theta) + \frac{\lambda}{2}\|\Theta\|_F^2$ converges to stationary points satisfying the KKT conditions. To complete the picture, we must link this convergence guarantee back to the gradient properties derived in Theorem C.5, which analyzed the unregularized population risk $\mathcal{R}(\Theta)$.

We now demonstrate that the regularization term $\frac{\lambda}{2}\|\Theta\|_F^2$ works in tandem with the data-driven suppression mechanism. Specifically, for any nonnegative direction $\Delta \geq 0$ in the $J_n$-channel, the directional derivative of the regularized objective satisfies:

$$D_{B_\ell}\mathcal{R}_\lambda(\Theta)[\Delta] = \underbrace{D_{B_\ell}\mathcal{R}(\Theta)[\Delta]}_{\text{Population Risk Gradient}} + \underbrace{\lambda\langle B_\ell, \Delta\rangle_F}_{\text{Regularization Penalty}}. \tag{29}$$

When the cross-component penalty dominates the within-component reward (as characterized in Theorem C.5), the population risk gradient is already nonnegative ($D_{B_\ell}\mathcal{R}(\Theta)[\Delta] \geq 0$). Since $\lambda > 0$ and $B_\ell \geq 0$, the regularization term is also nonnegative, and strictly positive whenever $B_\ell \neq 0$. Consequently, the regularization only strengthens the inequality, ensuring that the heuristic channel is suppressed at any stationary point.

**Corollary C.20** (Regularized KKT Points Suppress the Heuristic Channel). *Assume the setting of Assumption 4.8. Let $\lambda > 0$ and let $\Theta_\lambda^* = (A_\ell^*, B_\ell^*)_{\ell=1}^L$ be a KKT point of the regularized objective $\mathcal{R}_\lambda(\Theta)$ over the constraint set $\mathcal{C}$. Fix a layer $\ell \in \{1, \ldots, L\}$ and assume the Population-Level Dominance Condition (cf. Equation (7)) holds at $\Theta_\lambda^*$:*

$$\mathbb{E}\left[\sum_{R_{ij}=0} D_{ij}\right] > \mathbb{E}\left[\sum_{R_{ij}=1} \frac{1 - \phi_\epsilon(Z_{ij})}{\phi_\epsilon(Z_{ij})} D_{ij}\right], \tag{30}$$

*where $D = \frac{\partial Z}{\partial B_\ell}[B_\ell^*]$ is the Jacobian along the direction of the learned weights $B_\ell^*$. Then the heuristic channel is fully suppressed: $B_\ell^* = 0$.*

*Proof.* We proceed by contradiction. Suppose $B_\ell^* \neq 0$.

Since $\Theta_\lambda^*$ is a KKT point of $\mathcal{R}_\lambda$ over $\mathcal{C}$, the first-order optimality conditions for the non-negative parameter $B_\ell^*$ require

$$(\nabla_{B_\ell}\mathcal{R}(\Theta_\lambda^*) + \lambda B_\ell^*) \odot B_\ell^* = 0. \tag{31}$$

Summing over all entries (taking the Frobenius inner product with $B_\ell^*$), we obtain:

$$\langle\nabla_{B_\ell}\mathcal{R}(\Theta_\lambda^*), B_\ell^*\rangle_F + \lambda\|B_\ell^*\|_F^2 = 0. \tag{32}$$

We now analyze the population gradient term $\langle\nabla_{B_\ell}\mathcal{R}(\Theta_\lambda^*), B_\ell^*\rangle_F$. By Theorem C.5, this term decomposes into the difference between the expected penalty on disconnected graphs and the expected reward on connected graphs. By the Population-Level Dominance Condition assumed in the Corollary statement, the expected penalty strictly exceeds the expected reward. Therefore, the unregularized gradient contribution is strictly positive:

$$\langle\nabla_{B_\ell}\mathcal{R}(\Theta_\lambda^*), B_\ell^*\rangle_F > 0. \tag{33}$$

Intuitively, this means the data distribution itself is pushing the weights $B_\ell^*$ toward zero.

Substituting this into equation 32, we arrive at a contradiction:

$$\underbrace{\langle\nabla_{B_\ell}\mathcal{R}(\Theta_\lambda^*), B_\ell^*\rangle_F}_{>0} + \underbrace{\lambda\|B_\ell^*\|_F^2}_{>0} = 0. \tag{34}$$

Both terms on the left-hand side are strictly positive (the second term because $\lambda > 0$ and we assumed $B_\ell^* \neq 0$). Their sum cannot be zero. Thus, we must have $B_\ell^* = 0$. $\qquad\square$

*Remark* C.21 (Role of Regularization). The regularization parameter $\lambda > 0$ plays a dual role. First, it ensures coercivity (Lemma C.16), which is necessary to prove the existence of limit points for the training dynamics in Theorem 4.9. Second, as shown in Corollary C.20, it acts as a strict enforcer of suppression: even if the data-driven gradient were merely zero (a "tie" between penalty and reward), the regularization force $\lambda B_\ell^*$ would still drive the weights to zero via the stationarity condition.

**Corollary C.22** (Convergence to Algorithmic Solutions). *Under Assumption 4.8, let $\{\Theta^{(k)}\}_{k\geq 0}$ be the sequence generated by projected gradient descent on $\mathcal{R}_\lambda$ with step size $\eta \leq 1/L$ and initialization $\Theta^{(0)} \in \mathcal{C}$. Suppose that at every limit point $\Theta^*$, the Population-Level Dominance Condition from Corollary C.20 holds for all layers $\ell$.*

*Then every limit point satisfies $B_\ell^* = 0$ for all $\ell$, and consequently $W_\ell^* = A_\ell^* \otimes I_n$. The model at any limit point implements the algorithmic $I_n$-channel exclusively and reaches its theoretical capacity of $3^L$.*

*Proof.* Theorem 4.9 guarantees that every limit point $\Theta_\lambda^*$ satisfies the KKT conditions for $\mathcal{R}_\lambda$ over $\mathcal{C}$. Applying Corollary C.20 to each layer yields $B_\ell^* = 0$. With $B_\ell^* = 0$, the heuristic channel is eliminated. The capacity statement then follows from Lemma C.12, which establishes that when $B_1 = \cdots = B_L = 0$, the model output satisfies $Z_{ij} > 0$ only if $[A^{3^L}]_{ij} > 0$, achieving the tight capacity bound of Theorem 4.5. $\qquad\square$

# D  Additional Experiment Details and Results

## D.1  Experiment Details

**Standard Transformers.** When training 2-layer standard Transformers, we adopt the implementation from RoBERTa (Liu et al., 2019) with single-head per-layer and using normalized ReLU activation function as defined in Definition A.1. We use a hidden dimension of $d = 512$ to make sure the hidden size is not the blocker for expressivity. We trained on 1 Billion ER graphs with a batch size of 1000 and $10^6$ steps. Each graph is only seen by the model once to resembling the training regime of modern LLMs. We note that although 1 billion graphs sounds a lot but with $n = 20$ nodes, this is far from enumerating all possible graphs: there can be $2^{\binom{n}{2}}$ graphs if we don't consider graph isomorphism. When $n = 20$, this is about more than $10^{57}$ graphs in total, and 1 billion ($10^9$) is only a very small number of training instances. We train with AdamW optimizer with a learning rate of `1e-4` and weight decay of `1e-4` and a cosine learning rate decay.

**Disentangled Transformers.** For 1-layer Disentangled Transformers in Section 5, we train on a fixed set 4096 i.i.d. samples of ER($n = 8$) graphs and running standard *Gradient Descent* without any mini-batching. In this case, we have a learning rate of 0.1 with cosine learning rate decay. For 2-layer Disentangled Transformers, we train on the same set of 1 billion number of ER($n = 20$) graphs as with standard Transformers. For 3-layer models, we train on 1 billion number of ER($n = 64$) graphs. Both 2- and 3-layer models are trained with AdamW with a learning rate of `1e-3`. We would like to note that the hidden dimensions $d_\ell$ of Disentangled Transformers are fixed to be $d_\ell = 2^{\ell+1}n$ rather than a hyper-parameter (see Definition 4.1).

**Computing Energy Share of $I_n$/$J_n$ Channels.** In the experiments on 1-Layer Disentangled Transformers, we compute energy shares of the $A \otimes I_n$ and $B \otimes J_n$ within $\|W\|_F^2$. Here is the formalized versions. We consider the noisy decomposition $W = \hat{A} \otimes I_n + \hat{B} \otimes J_n + W_\epsilon$, where $W_\epsilon$ is the projection error term. We define Frobenius-norm energy share on the $I_n$ channel as

$$\mathsf{EnergyShare}(\hat{A} \otimes I_n, W) = \frac{\langle W, \hat{A} \otimes I_n\rangle}{\|W\|_F^2} = \frac{\|\hat{A} \otimes I_n\|_F^2 + \langle \hat{A} \otimes I_n, \hat{B} \otimes J_n\rangle + \langle \hat{A} \otimes I_n, W_\epsilon\rangle}{\|W\|_F^2},$$

and by symmetry, the $J_n$-channel share is

$$\mathsf{EnergyShare}(\hat{B} \otimes J_n, W) = \frac{\langle W, \hat{B} \otimes J_n\rangle}{\|W\|_F^2} = \frac{\|\hat{B} \otimes J_n\|_F^2 + \langle \hat{B} \otimes J_n, \hat{A} \otimes I_n\rangle + \langle \hat{B} \otimes J_n, W_\epsilon\rangle}{\|W\|_F^2}.$$

This is a well-designed quantity because expanding $\|W\|_F^2$ gives $\langle W, \hat{A} \otimes I_n + \hat{B} \otimes J_n + W_\epsilon\rangle$, and the $I$/$J$-channels' energy shares will sum to one when the projection error $W_\epsilon$ converges to zero.

## D.2  Additional Experiments on Disentangled and Standard Transformers

In Figure 7, we show the training dynamics of a 3-Layer Disentangled Transformer. In Figure 8, we show the learned weights by Disentangled Transformers.

In Figure 8, we show that the trained 2-layer and 3-layer converge to weight spaces $W_\ell = A_\ell \otimes I_n + B_\ell \otimes J_n$ in the particular form echoing Theorem 4.7.

In Figure 9, we show that the capacity theorems (Theorem 4.5) also transfer to standard 2-layer Transformer models.

In Figure 10, we show that when evaluated in 2Clique dataset, the one trained with the right data generalize better.

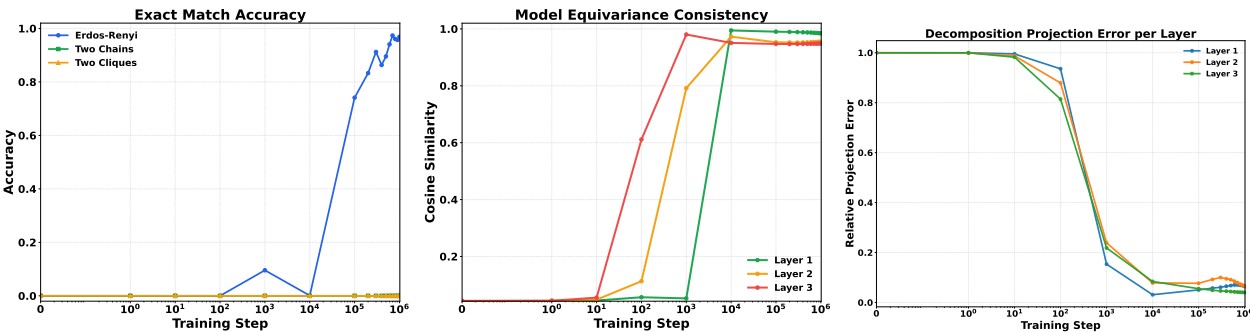

*Figure 7.* We plot the model behavior of a 3-Layer Disentangled Transformer model trained on $\mathsf{ER}(n = 64)$ graphs. They also quickly pick up almost *layer-wise equivariant* properties (measured by Eqn. 4). All layers show very small projection error onto the $A \otimes I_n + B \otimes J_n$ decomposition, resonating our theoretical claims in Theorem 4.7.

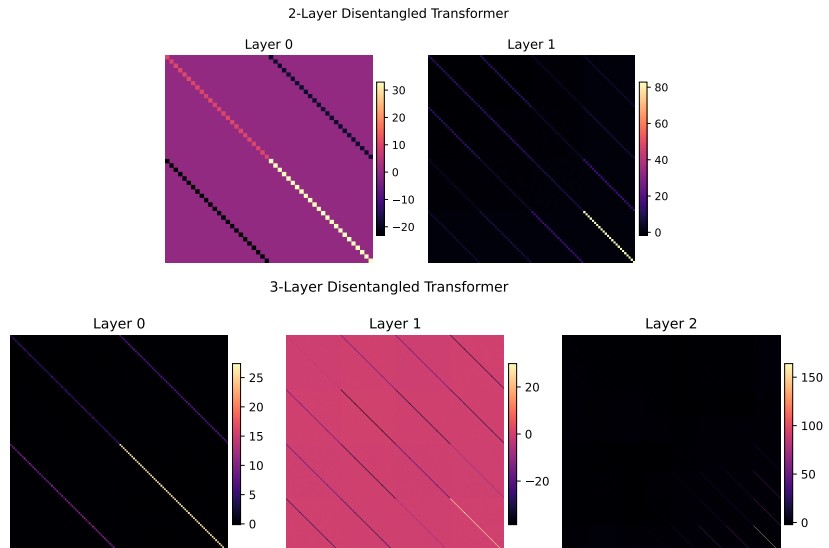

*Figure 8.* Here we visualize the weights $W_\ell$ learned by a 2-Layer and 3-Layer Disentangled Transformer respectively. All models are randomly initialized **without** any restriction on parameterization. Resonating Theorem 4.7, they all converge to a form of $W_\ell = A_\ell \otimes I_n + B_\ell \otimes J_n$.

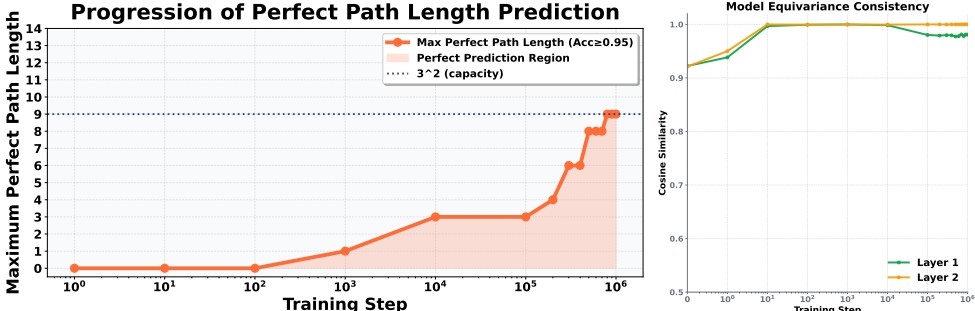

*Figure 9.* (**left**) Standard Transformers models studied in §3.3 also hit its capacity wall at $3^L$, showing that our theoretical results transfer beyond the theoretical simplification of Disentangled Transformers. (**right**) Standard Transformer models also learn an almost layer-wise equivariant solution measured by Eqn. 4.

## D.3  Scaling Effects of Diameter and Capacity

In Figure 11, we study the effects of varying the restrictions on graph diameters and edge probability $p$ in generating $\mathsf{ER}(n, p)$ graphs, and we find that length generalization is particularly hard and at-capacity graphs are important as well. In

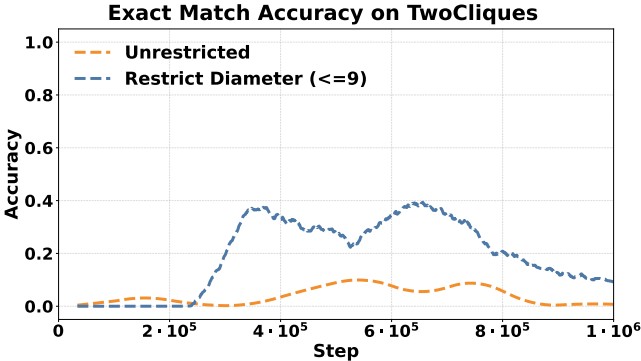

*Figure 10.* Under the same setup as Fig. 6, when tested on 2Clique graphs, the one trained with *the right data* is able to generalize better.

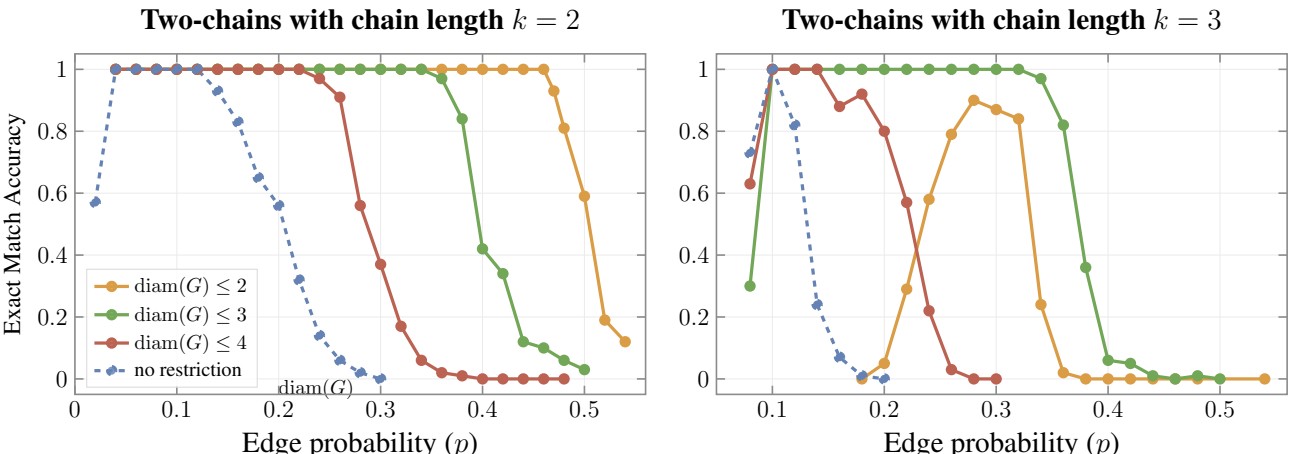

*Figure 11.* With 1-layer Disentangled Transformers with capacity $\mathsf{Cap} = 3$ following Theorem 4.5, we vary $d$ such that we restrict our training graphs to have $\mathrm{diam}(G) \leq d$. We also vary the edge probability of our training distribution $\mathsf{ER}(n = 8, p = \cdot)$ for generality. We test on $\mathsf{2Chain}(n = 8, k = \cdot)$ graphs with $k = 2$ or 3 and show the exact match accuracy on configurations where the accuracy is non-zero for readability. We find if the training $d \leq \mathsf{Cap}$, models still learns the algorithmic solution up to problem size $d$ (see $d = 2, k = 2$ case on the left in orange) but *fails to length generalize* (see $d = 2, k = 3$ in orange on the right). On the other hand, if the training $d > \mathsf{Cap}$, model struggles to learn the algorithmic solution (see $d = 4$ cases in red on both $k = 2$ or 3). The best case overall is when setting $d = \mathsf{Cap}$, i.e., preventing the model from seeing beyond-capacity samples but still preserving at-capacity samples for better generalization. As shown in the green lines, with $d = 3$, model achieves balanced testing accuracy on both $k = 2$ and 3.

Figure 12, we study the effects of at-capacity graphs. Without at-capacity graphs, models struggle to learn the algorithmic solution as well.

# E   Additional Related Work

**Mechanistic Interpretability of Transformers.** A growing body of work reverse-engineers the *algorithmic circuits* that Transformers learn for tasks like copying, induction, and reasoning (Elhage et al., 2021; Olsson et al., 2022; Wang et al., 2023; Brinkmann et al., 2024). These can range from Fourier-style circuits for modular addition (Nanda et al., 2023; Zhou et al., 2024c) to Newton-like updates for in-context linear regression (Fu et al., 2024a). Researchers validate hypotheses by compiling programs into model weights (Lindner et al., 2023), decompiling models into code (Friedman et al., 2023), and using causal interventions to localize function (Chan et al., 2022; Meng et al., 2022; Yao et al., 2024; Chang et al., 2024). Theoretical work on inductive biases, like a preference for low-sensitivity functions, helps explain why models often favor robust heuristics over exact algorithms (Vasudeva et al., 2025).

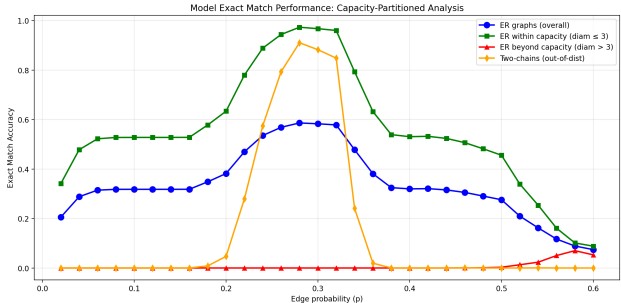

*(a)* When training 1-layer Disentangled Transformers, instead of restricting training graphs to have diameter at most 3, we restrict $\mathrm{diam}(\mathbf{G}) \leq \mathbf{2}$ and vary the edge probability in $\mathsf{ER}(n = 8, p = p)$ training distribution. Measured by exact-match accuracy, restricting $\mathrm{diam}(G) \leq 2$ makes the models unable to generalize as well, indicating the importance of **at-capacity graphs** ($\mathrm{diam}(G) = 3$).

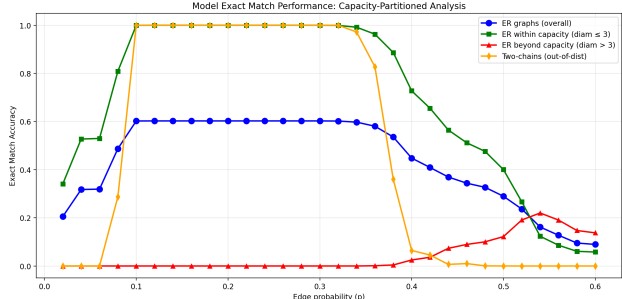

*(b)* When restricting $\mathrm{diam}(\mathbf{G}) \leq \mathbf{3}$, with reasonable $p \in [0.1, 0.32]$, the 1-layer Disentangled Transformer can learn the algorithmic channel.

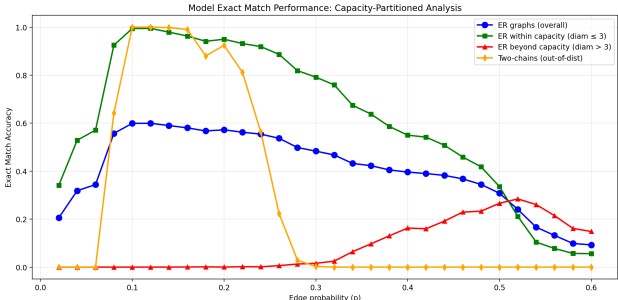

*(c)* When restricting $\mathrm{diam}(\mathbf{G}) \leq \mathbf{4}$, allowing some beyond-capacity graphs, the 1-layer Disentangled Transformer struggles to learn the algorithmic channel, and starts to rely on the heuristic $J_n$-channel to make predictions on beyond-capacity graphs (red lines).

*Figure 12.* Effects of **at-capacity graphs** ($\mathrm{diam}(G) = 3^L$) for $L = 1$. Without at-capacity graphs, models struggle to learn the algorithmic solution. With beyond-capacity graphs, models weight heuristics too heavily. In short, models not only need most graphs within capacity but also require at-capacity graphs to learn algorithms over heuristics.

