# OpenReview forum: "Transformers Provably Learn Algorithmic Solutions for Graph Connectivity, But Only with the Right Data"
_ICML.cc/2026/Conference — ICML 2026 regular_

### Official Review · Reviewer_Mzjd · 2026-03-11

**Soundness:** 4
**Presentation:** 4
**Significance:** 4
**Originality:** 4
**Overall Recommendation:** 5
**Confidence:** 1

**Summary:**

The authors address the relation between the complexity of a transformer's architecture (measured in terms of the number of its layers) and its capacity to learn mathematical concepts. In particular, they focus on the problem of checking the connectivity of a given graph with a disentangled transformer. On the theoretical side, they show that an $L$-layer disentangled transformer suffices to correctly detect all graphs of diameter at most $3^L$, and prove that this bound is tight, i.e., there exists a graph of diameter $3^L+1$ for which no $L$-layer disentangled transformer learns correct connectivities. They show that under reasonable assumptions, learned weights decompose into two channels: the algorithmic channel attempts to learn the concept of connectivity and is favored by all training graphs whose diameter is at most $3^L$; the heuristic channel on the other hand learns global statistics (e.g. detecting connectivity based on vertex degrees, which yields incorrect predictions for instance in a cluster graph that consists of more than 1 component) and is favored by training graphs whose diameter is larger than $3^L$. This two-channel-story is backed up with some experiments that show that altering the training distribution such that either "small" diameter graphs or "big" diameter graphs dominate the data set pushes the model towards either learning the correct mathematical concept or a global heuristic.

**Compliance With Llm Reviewing Policy:**

Affirmed.

**Final Justification:**

I keep my original positive score as the paper seems sound and relevant. However, this should be weighted with my low confidence score as the work is not in my area.

**Key Questions For Authors:**

I recall here the questions that I raised above:

1. How do the results generalize to other mathematical problems? In particular,

1a. How general is the two-channel observation?

1b. Do other problems admit tight characterizations in terms of the architectures that are needed to learn them?

2. What are the major challenges that arise when transferring the theoretical results from the restricted disentangled transformer setting to general transformers?

**Limitations:**

yes

**Strengths And Weaknesses:**

The studied problem seems natural and interesting; the paper reads very well, and is well-motivated by previous work in both theory and practice. I could imagine it to attract the interest of theoreticians and practitioners alike.

Methodology-wise, the authors explain their restriction to the problem of graph connectivity in a very transparent way, stating that this problem admits a clear algorithmic solution, is learnable by certain transformers (although the previously existing bound is weaker than the one the authors propose), and admits simple heuristic strategies. While I see no issue with this restriction, I would have appreciated a paragraph that reflects on possible generalizations: Do the authors expect to detect the two-channel-property in all problems with the above properties? What is known about the necessary transformer complexity of other mathematical problems? In particular, has their architecture been characterized as tightly as in the graph connectivity case?

Moreover, since the experiments suggest that the proposed results generalize to arbitrary transformers, I wondered whether the authors also considered them in the theoretical setting. What would be the major challenges in lifting the theory from disentangled transformers to general transformers?

That being said, the submission is not in my area, so I can only assess its impact and soundness on a very high level.

---

> ### Author Rebuttal · Authors · 2026-03-31
>
> We thank the reviewer for finding the problem natural and interesting, and for noting that the paper is well-motivated and could attract interest from both theoreticians and practitioners. We address the questions below.
>
> **W1/Q1 (Two-channel generality and tight bounds):** We expect the two-channel phenomenon to appear in other tasks where (i) the task admits a clean algorithmic solution, (ii) simple heuristics correlate with the correct answer on typical data, and (iii) the model architecture has a finite ``capacity’’ determined by depth. Graph connectivity satisfies all three, but so should other problems: for instance, multi-digit addition has sequential carries as the "algorithmic" path and digit-level correlations as a potential "heuristic" shortcut. On tight bounds: to our knowledge, ours is among the first exact (non-asymptotic) capacity bounds for a specific algorithmic task on Transformers. Our methodology (analyze simplified architecture for exact bounds, validate on standard architectures) could apply to tasks like sorting, shortest paths, or parity. The challenge is identifying the right simplified architecture that admits exact analysis while remaining predictive. While our theoretical results are specific to graph connectivity and Disentangled Transformer, the broader framework (identify capacity $\to$ decompose channels $\to$ calibrate training data) should generalize. We will add this discussion in Section 6.
>
> **W2/Q2 (Extending to general Transformers):** Three main challenges: (1) residual connections add features in a shared space rather than concatenating, so the clean channel separation breaks down; (2) LayerNorm introduces nonlinear rescaling that couples the algorithmic and heuristic channels; (3) softmax attention normalizes globally rather than applying ReLU elementwise, changing propagation dynamics. Despite all three, Figures 6, 9, 10 show our capacity bound and data lever transfer to standard Transformers consistently. We do not enforce nonnegativity or any structural constraints during training of standard Transformers, yet our predictions hold. Formal analysis for standard Transformers remains an important open problem, but the empirical agreement across all our experiments is consistent with the theory.

---

> > ### Author Rebuttal · Reviewer_Mzjd · 2026-04-01
> >
> > I thank the authors for addressing my questions, there are no follow-up questions from my side.

---

> > > ### Author Response · Authors · 2026-04-07
> > >
> > > Thank you for the positive assessment and for confirming our rebuttal fully resolved your questions. We will add the promised discussion on generalizability in the final version.

---

### Official Review · Reviewer_8r5P · 2026-03-11

**Soundness:** 1
**Presentation:** 3
**Significance:** 2
**Originality:** 3
**Overall Recommendation:** 3
**Confidence:** 3

**Summary:**

The goal of this paper is to determine whether transformer models can learn a connectivity algorithm, rather than merely heuristic information correlated with connectivity on the training sets. Using a simplified transformer variant known as a disentangled transformer, the authors develop a fairly elaborate theory that distinguishes between "heuristic" and "algorithmic" channels and, to some extent, empirically verify their results.

**Compliance With Llm Reviewing Policy:**

Affirmed.

**Key Questions For Authors:**

I did not find information on the edge probability of the graphs you use in your larger experiments. Please provide it.

**Limitations:**

yes

**Strengths And Weaknesses:**

The goal of the paper, understanding the capability of transformer models to learn algorithms, is a laudable one. I appreciate that the authors root their approach in careful theoretical considerations. Yet I am not fully convinced by the paper in two ways.

The first is the basic question of what it would mean to learn an algorithm. To me, it is a fundamental property of a connectivity algorithm to generalise to large graphs without changing the architecture, and this requires some form of recursion in the architecture. Indeed, recursion (or iteration) is a fundamental algorithmic principle, so I would not see what is studied here as algorithmic.

The second point is more down-to-earth: I find the experiments very unconvincing due to the very restrictive and frankly, bad choice of training data. Trying to learn a connectivity algorithm on Erdös-Rényi random graphs is simply not a good idea, because of their connectivity structure. Also, the graphs considered are fairly small; to get meaningful results, I would want to see graphs of at least a couple of hundred vertices. In particular, the ER graphs on 20 vertices with an edge probability of 0.08 used in the preliminary study seem ill-suited for learning a connectivity algorithm. (In expectation, these graphs have 15 edges on 20 vertices. Have you looked at them?)

I do think the approach has potential to be very interesting, but it needs a more convincing empirical evaluation.

---

> ### Author Rebuttal · Authors · 2026-03-31
>
> We thank the reviewer for finding the goal of understanding Transformer algorithmic capability "laudable" and for appreciating that our approach is rooted in careful theoretical considerations. We address the concerns below.
>
> **W1 ("Algorithm" and recursion):** We appreciate this point and agree that generalizing to arbitrary-size graphs without changing the architecture is an important goal. Our model does not achieve this: it solves connectivity up to diameter $3^L$ for a fixed $L$. What it does achieve is provably implementing the matrix powering algorithm $\sum_{j=0}^{3^L} \alpha_j A^j$ with positive $\alpha_j$, which is the correct algorithm within its capacity. We use "algorithmic" to contrast with the degree-counting heuristic (the $J$-channel), which computes a fundamentally different function. The paper's contribution is characterizing when training selects one over the other. We note that studying what fixed-depth Transformers can compute is standard in this area (Merrill & Sabharwal 2023, 2025; Sanford et al. 2024), and we believe the question of what training dynamics select within the expressive class is complementary to the question of scaling to arbitrary sizes. We will clarify this distinction in revision.
>
> **W2/Q (ER graphs, choice of $p$, and graph size):** We chose $p$ for each $n$ so that $\Pr[R_{i,j} = 1] \approx 0.5$ for a random node pair $(i,j)$. This gives us balanced labels in the training data: roughly half the node pairs are connected and half are not, which is the maximally informative regime for learning connectivity. Here are the values we used:
>
> | $n$ | $p$ | $\Pr[i \sim j]$ | Used in |
> |---|---|---|---|
> | 8 | 0.20 | $\approx 0.50$ | Fig 3, 4, 11, 12 |
> | 20 | 0.08 | $\approx 0.43$ | Fig 1, 6, 9, 10 |
> | 24 | 0.07 | $\approx 0.46$ | Fig 2 (top) |
> | 64 | 0.027 | $\approx 0.51$ | Fig 2 (bottom), Fig 7, 8 |
>
> We will state these values and the selection criterion explicitly in revision. On connectivity structure: by the Erdős-Rényi connectivity threshold (Bollobás, 2001), $G(n,p)$ is connected w.h.p. when $p > \frac{\ln n}{n}$, and a giant component of $\Theta(n)$ nodes emerges when $np > 1$. All our choices sit near this transition, producing a nontrivial mix of connected and disconnected graphs with varying diameters. This is exactly what our theory requires (Assumption 4.8). On graph size: our contribution concerns diameter vs. depth, not absolute $n$. A 3-layer model on $n=64$ with capacity $3^3=27$ tests the same mechanism regardless of $n$. We have also run 4-layer standard Transformer experiments on $n=192$ and the $3^L$ capacity bound continues to hold. We are running larger-scale Disentangled Transformer experiments and will include results in the discussion period if completed in time, or in the revision if it takes longer.
>
> We want to emphasize that our theory is not ER-specific. The capacity bound ($3^L$) holds for any graph. The data lever prescription (restrict to within-capacity graphs) is distribution-agnostic. The training dynamics analysis (Theorems C.4, C.5, C.9) only requires that the data distribution has positive probability of disconnected graphs (Assumption 4.8), which holds for any random graph model near its connectivity threshold. We test on 2Chain and 2Clique (Figures 1, 6, 10), which have completely different topology from ER graphs: deterministic, regular, chain/clique geometry. Our data lever trained on filtered ER graphs generalizes to these. If we had fit ER-specific correlations, this transfer would not happen. We would welcome further discussion on our main technical contributions and are happy to clarify any aspect.
>
>
> **Reference**
>
> Bollobás B. _Random Graphs. 2nd ed._ Cambridge University Press; 2001.

---

> > ### Author Rebuttal · Reviewer_8r5P · 2026-04-01
> >
> > Thanks for your responses. I would still maintain that ER random graphs are not the ideal testbed for connectivity algorithms, and the fact remains that the graphs considered are fairly small.
> >
> > I'm afraid my overall assessment of the paper remains unchanged: this is an interesting direction, but I don't find the paper strong enough for a top conference like ICML.

---

> > > ### Author Response · Authors · 2026-04-07
> > >
> > > Thank you for acknowledging that our rebuttal resolved your stated concerns. Since the rebuttal, we have conducted additional experiments addressing your remaining reservations: we verified the $3^L$ capacity bound on standard Transformers trained on Stochastic Block Model (SBM) graphs (a structurally different graph distribution compared to Erdos-Renyi graphs). We trained both for n=64 with 3 layers and n=24 with 2 layers. We will continue train on SBM graphs for the data lever experiments and potentially extend to more graph families and larger graphs. We will include these in the revision. Again, we thank you sincerely for your constructive feedback and suggestions.

---

### Official Review · Reviewer_2n7i · 2026-03-12

**Soundness:** 4
**Presentation:** 3
**Significance:** 3
**Originality:** 3
**Overall Recommendation:** 5
**Confidence:** 4

**Summary:**

This paper investigates why Transformers tend to learn brittle shortcuts rather than generalizable algorithmic logic when tackling algorithmic reasoning tasks. Using graph connectivity as a case study, the authors analyze a simplified architecture — the Disentangled Transformer — and establish four core contributions: (1) a tight capacity threshold showing that an L-layer model can only solve connectivity for graphs with diameter ≤ 3L; (2) a dual-channel decomposition of learned weights into an algorithmic I-channel (performing matrix exponentiation) and a heuristic J-channel; (3) a characterization of training dynamics showing that data distribution determines which channel dominates; and (4) a practical "data lever" strategy — restricting training samples to those within the 3L capacity bound — that steers standard Transformers toward robust, OOD-generalizable algorithmic solutions.

**Compliance With Llm Reviewing Policy:**

Affirmed.

**Final Justification:**

My final recommendation is positive. The rebuttal satisfactorily addressed my main concerns and strengthened my confidence in the paper, so I raised my score by one point.

**Key Questions For Authors:**

1. In Theorem 4.5, the requirement n = Ω(3L) is clarified only in Footnote 2. I recommend stating the explicit node count condition (e.g., n ≥ (7/3)·3L + 2) directly in the theorem statement to improve readability.
2. The mathematical formulation of "suppression" and "driving" forces in the Phase 2 training dynamics (Equation 6) is dense. A brief intuitive explanation of when Bℓ → 0 in the main text would help readers without a strong theory background.
3. The color coding in Figures 11 and 12 may be indistinguishable in grayscale printing. I recommend supplementing with distinct line styles.

**Limitations:**

yes

**Strengths And Weaknesses:**

S1. The paper provides a non-asymptotic, exact constant bound (3L) rather than the asymptotic O(log n) results typical in this area. This level of precision is rare in deep learning theory and constitutes a meaningful technical contribution.

S2. The dual-channel decomposition offers an interpretable lens for understanding Transformer internals, and directly explains the phenomenon of "reasoning brittleness" when models encounter out-of-capacity samples.

S3. Theoretical results derived for the Disentangled Transformer are empirically validated on standard Transformers, strengthening the claim of generality beyond the simplified architecture.

S4. The "data lever" insight has direct practical implications: it provides principled guidance for dataset curation, specifically that training difficulty should be calibrated to model depth.

W1. Several core theorems (e.g., Theorems 4.5 and 4.7) assume non-negative model weights. However, practically trained Transformers contain negative weights and use LayerNorm and different activation functions. The paper does not sufficiently discuss whether the 3L bound shifts under relaxed non-negativity, nor does it provide a sensitivity analysis for negative weights. I recommend the authors add such a discussion in the main text or appendix.

W2. The theoretical analysis is primarily built on the Disentangled Transformer, which uses feature concatenation rather than the residual summation of standard Transformers. Although approximate equivalence is argued, feature accumulation in residual streams may be more complex than concatenation implies. I suggest the authors examine how varying hidden dimension affects the 3L bound, to confirm whether this constant is truly independent of model width.

W3. The conclusions are tightly coupled to graph connectivity as a task. The paper would benefit from a brief discussion of how the notions of "diameter" and "capacity" might analogize to other reasoning tasks (e.g., arithmetic reasoning or logical inference), which would broaden the paper's impact.

---

> ### Author Rebuttal · Authors · 2026-03-31
>
> We thank the reviewer for appreciating the exact $3^L$ bound as a rare non-asymptotic result in deep learning theory (S1), the interpretability of the dual-channel decomposition (S2), and the empirical validation on standard Transformers and the data lever's transfer (S3, S4). We address the reviewer's concerns below.
>
> **W1 (Nonneg / LayerNorm / activations):** We do **not** enforce nonnegativity during training. All models use unconstrained weights. Nonnegativity is a proof technique only. Despite this, predictions hold: standard Transformers match the $3^L$ bound (Figure 9), the $A \otimes I + B \otimes J$ decomposition emerges naturally (Figures 3, 7, 8), and the data lever transfers to standard Transformers (Figure 6).
>
> What could change with negative weights, LayerNorm, or softmax? Negative weights could break the monotonicity in Lemma C.6, potentially enabling strategies outside $A \otimes I + B \otimes J$. LayerNorm couples channels via nonlinear rescaling. Softmax normalizes scores globally rather than applying ReLU elementwise. All three are present in standard Transformers, yet predictions hold (Figures 6, 9, 10). We will add a dedicated paragraph discussing these limitations and why the empirical transfer is informative but not a formal guarantee.
>
> **W2 (Disentangled Transformer concat vs. standard residual; hidden dim):** In Disentangled Transformer, layers append features so contributions are cleanly separated. In standard Transformers, residual connections add in a shared $d$-dimensional space, creating potential interference. Nichani et al. (2024) show any standard attention-only Transformer can be re-expressed as a Disentangled Transformer by specializing attention heads to implement concatenation. More importantly, the $3^L$ capacity bound is a sharp quantitative prediction: if the Disentangled Transformer-to-standard transfer were unreliable, we would expect standard Transformers to hit their capacity wall at a different value, but Figure 9 shows they land at exactly $3^2 = 9$. Our data lever also transfers (Figure 6). Closing this gap formally remains open, but the quantitative agreement is striking. On hidden dimension: in Disentangled Transformer, this is not a free parameter. It is solely determined by the input dimension, growing as $d_\ell = 2^{\ell+1} n$ at layer $\ell$ (Definition 4.1). The $3^L$ bound reflects how far information propagates through attention (each layer triples a node's reach), which depends more on depth than width.
>
> **W3 (Other tasks):** "Diameter" for connectivity is analogous to "reasoning depth" in sequential tasks. Multi-digit addition requires $O(\text{digits})$ sequential carries; logical inference depends on proof depth. The $3^L$ constant is task-specific, but our framework (identify capacity $\to$ decompose channels $\to$ calibrate training data) should generalize. We will add this discussion in §6.
>
> **Q1 (Theorem 4.5):** We will move $n \geq \frac{7}{3} \cdot 3^L + 2$ into the theorem statement.
>
>
>  **Q2 (Phase 2 intuition):** We will add a plain-language summary. The $J$-channel faces opposing forces: disconnected graphs penalize it (false positives push $B_\ell \to 0$), while connected beyond-capacity graphs reward it (the $J$-channel is the only way to reach distant pairs). When within-capacity graphs dominate, the penalty wins.
>
>
> **Q3 (Figures 11/12):** We will add distinct line styles for grayscale.

---

> > ### Author Rebuttal · Reviewer_2n7i · 2026-04-03
> >
> > My final recommendation is positive. The rebuttal satisfactorily addressed my main concerns and strengthened my confidence in the paper, so I raised my score by one point.

---

> > > ### Author Response · Authors · 2026-04-07
> > >
> > > Thank you for the positive assessment and for raising your score!
> > >
> > > We are glad our responses on the nonnegative-weight assumption, concatenation vs. residual distinction, and generalizability were convincing. All suggested presentation improvements will be included in the revision.

---

### Official Review · Reviewer_tgX8 · 2026-03-13

**Soundness:** 4
**Presentation:** 3
**Significance:** 3
**Originality:** 3
**Overall Recommendation:** 4
**Confidence:** 4

**Summary:**

This paper proves a tight diameter-based expressivity result for a simplified Disentangled Transformer, showing that an L-layer model can solve connectivity up to diameter 3^L but not beyond under a nonnegative-weight assumption. The more distinctive contribution is then to move beyond bare expressivity and analyze what training actually selects within that expressive class: the paper argues that within-capacity data encourages an algorithmic matrix-powering solution, while beyond-capacity data pushes the model toward a simpler heuristic channel. It then validates this message empirically by showing that restricting training data to the right diameter regime improves out-of-distribution generalization for both the simplified architecture and standard Transformers.

**Compliance With Llm Reviewing Policy:**

Affirmed.

**Final Justification:**

The authors' rebuttal meaningfully addressed my concerns.

**Key Questions For Authors:**

1. How should readers interpret the chain of assumptions in the dynamics analysis relative to standard end-to-end Transformer training? Which assumptions are mainly technical devices, and which are believed to capture the real mechanism?

2. How should readers interpret the nonnegative-weight assumption relative to the unconstrained models used in practice? Is there empirical evidence that training stays close to the regime analyzed by the theory?

3. Since Erdos-Renyi graphs connectivity is highly concentrated, how sensitive is the central message to the choice of training distribution? Would the same conclusions hold on graph families with different connectivity phase behavior or more heterogeneous geometry?

4. Could the paper include more diagnostic experiments or visualizations showing the transition between heuristic and algorithmic behavior?

**Limitations:**

The paper would benefit from a more explicit limitations discussion. First, the most interesting contribution is the dynamics/message side, but that theory is not yet fully end-to-end and depends on several intermediate assumptions, so the explanatory scope should be stated carefully. Second, the core theoretical analysis still relies on a nonnegative-weight regime that is cleaner than practical training. Third, the empirical support, while useful, is still not especially broad: the paper does sweep aspects of the Erdos-Renyi setting, but broader graph families and more diagnostic visualizations would strengthen the case. These do not undermine the paper's message, but they do limit how broadly one can currently generalize it.

**Strengths And Weaknesses:**

## Strengths

**S1. The task and overall message are highly relevant.**
The paper tackles a question that is central in the current community: why models that are expressively capable of representing an algorithm still fail to actually learn it. This is timely and interesting, especially in the broader context of latent reasoning and algorithmic generalization.

**S2. The paper goes beyond pure expressivity analysis.**
The expressivity side by itself is not the most novel part, since prior work already established strong depth/connectivity phenomena. A real strength here is that the paper tries to analyze learning dynamics and explain when training selects an algorithmic solution versus a heuristic one.

**S3. The theory-to-experiment story is clear.**
The paper turns its analysis into a concrete data-design prescription and then tests that prescription empirically. That gives the work a clearer practical message than a paper that only proves an expressivity theorem.

**S4. The central message is interesting even beyond this exact task.**
The idea that failure can come not from lack of expressive power, but from the training distribution steering optimization toward a shortcut, is a useful and broadly relevant message for the field.

## Weaknesses

**W1. Novelty on the expressivity side.**
The expressivity result is clean, but the broader picture that logarithmic-depth Transformer-like models can represent connectivity-like algorithms is not entirely new. The paper's more distinctive contribution is the dynamics story, and it would be better to present the novelty with that emphasis.

**W2. The dynamics analysis is not fully end-to-end and relies on several intermediate assumptions.**
The most interesting part of the paper is the learning-dynamics account, but that account is established only under a chain of intermediate assumptions and reductions rather than as a full end-to-end explanation of standard training. This limits how directly one can interpret the theory as describing practical Transformer optimization.

**W3. The core theory depends on a nonnegative-weight assumption that is not enforced in practice.**
This is the main conceptual gap in the paper. The theory is clean in the constrained regime, but the practical models are trained without that constraint, so the bridge from theorem to standard optimization is not fully explained.

**W4. The experimental picture could be broader and more diagnostic.**
The experiments support the main message, but they are still relatively limited. The paper does vary aspects of the Erdos-Renyi training distribution, which is useful, but much of the empirical story remains centered on that family. It would be helpful to include a wider range of graph distributions or more diagnostic visualizations of the transition between algorithmic and heuristic regimes.

---

> ### Author Rebuttal · Authors · 2026-03-31
>
> We thank the reviewer for recognizing that our task and message are highly relevant (S1), that  our work goes beyond pure expressivity (S2), that the theory-to-experiment story is clear (S3), and that the central message about training distribution steering optimization is broadly relevant (S4). We address the weaknesses below.
>
> **W1 (Expressivity novelty):** Agreed. We will reframe to foreground the dynamics contribution.
>
> **W2/Q1 (Dynamics assumptions):** Our analysis is modular, with each component independently justified:
> - *Parameterization (Theorem 4.7)*: Our choice of parameterization arises from empirical observations. Figures 3, 7, 8 show that unconstrained models converge to this form on their own.
> - *KKT stationarity (Theorem 4.9)*: We fully proved this claim in Appendix C.5 via standard projected GD convergence (Bertsekas 1997; Beck 2017).
> - The dynamics story is centered around Theorem C.5. Figure 5 validates it directly: the algorithmic channel share tracks the within-capacity proportion continuously.
> We note that modularity is intentional here, so that if any assumption is later relaxed (e.g., nonnegativity, see below), the rest carries through unchanged. We will make this structure clearer in revision.
>
>
>  **W3/Q2 (Nonnegative weights):** We want to stress that we do not enforce nonnegativity in training and all models use fully unconstrained weights, so nonnegativity appears only in the proofs. Despite this, our results, proven under the assumption of nonnegative weights, hold across the full pipeline: the story on both capacity/expressivity (Figure 9) and the "data lever" (Figure 6) apply to Disentangled Transformers with unconstrained weights, as well as to standard Transformers. Furthermore, the $A \otimes I + B \otimes J$ decomposition emerges on its own without enforcement (Figures 3, 7, 8; projection error $\to 0$). We will add an explicit discussion of these observations and the remaining gaps between the nonnegative proof regime and unconstrained practice.
>
>
>  **W4/Q4 (Diagnostics):** For Disentangled Transformer, we can directly measure the algorithmic-vs-heuristic transition through the I/J channel energy shares in the weights (Figures 3, 4, 5, 12). For standard Transformers, this kind of direct weight-level diagnosis is harder because the residual stream mixes contributions from both channels. Our current diagnostic for standard Transformers is indirect: we observe whether OOD accuracy improves under the data lever (Figures 6, 10), which tells us the model shifted toward the algorithmic solution but does not let us visualize the internal transition as cleanly as we can for Disentangled Transformer. We acknowledge this gap and will discuss it in revision.
>
> **Q3 (ER distribution sensitivity):** We want to emphasize that our theory is not ER-specific. The capacity bound ($3^L$) holds for any graph, the data lever is distribution-agnostic, and the training dynamics analysis (Theorems C.5, C.9) only requires positive probability of disconnected graphs (Assumption 4.8). While we train on ER graphs, our OOD evaluation on 2Chain and 2Clique already tests transfer to structurally very different graph families. We also vary the ER edge probability systematically in Figures 11 and 12, and sweep the beyond-capacity proportion continuously in Figure 5. Training on non-ER families would further strengthen the empirical case, and we plan to explore this in revision.
>
> **Limitations:** We will add an explicit limitations paragraph covering the three points above: modular dynamics analysis with empirical bridges, nonnegativity as proof technique only, and scope of experiments.

---

> > ### Author Rebuttal · Reviewer_tgX8 · 2026-04-04
> >
> > I thank the authors for addressing my questions. I raise my overall score to 4.

---

> > > ### Author Response · Authors · 2026-04-07
> > >
> > > Thank you for confirming that our rebuttal fully resolved your concerns and for raising your score. We will incorporate your valuable suggestions in the revision.

---

### Decision · Program_Chairs · 2026-04-30

**Decision:**

Accept (regular)

**Comment:**

The paper aims to understand when transformers learn an algorithmic solution versus a heuristic shortcut for graph connectivity.

In general, reviewers agreed that the problem is somewhat interesting. In particular, the $3^L$
 capacity bound, the channel decomposition, and the resulting data-design insight were viewed as meaningful contributions. Multiple reviewers also found the interaction between theory and empirical study convincing.

One reviewer was not convinced by the empirical setting, arguing that using small Erdős--Rényi graphs is not convincing enough, and criticized the fixed-depth setting. Some reviewers also noted that the theoretical dynamics analysis is modular rather than fully end-to-end, and that the nonnegative-weight assumption creates a gap between the theoretical assumptions and standard unconstrained training.

In summary, the discussion supports acceptance. The paper has some limitations in the empirical study; the core contribution is somewhat interesting, clearly presented, and likely to be of interest to the community. I would recommend acceptance. I would enourage the authors to impriove on the emperical study.